# Second-Order Min-Max Optimization with Lazy Hessians

**Lesi Chen** [*]
IIIS, Tsinghua University & Shanghai Qizhi Institute
chenlc23@mails.tsinghua.edu.cn

**Chengchang Liu** [*]
The Chinese University of Hong Kong
7liuchengchang@gmail.com

**Jingzhao Zhang** [†]
IIIS, Tsinghua University & Shanghai Qizhi Institute & Shanghai AI Lab
jingzhaoz@mail.tsinghua.edu.cn

## Abstract

This paper studies second-order methods for convex-concave minimax optimization. Monteiro & Svaiter (2012) proposed a method to solve the problem with an optimal iteration complexity of $\mathcal{O}(\epsilon^{-3/2})$ to find an $\epsilon$-saddle point. However, it is unclear whether the computational complexity, $\mathcal{O}((N + d^2)d\epsilon^{-2/3})$, can be improved. In the above, we follow Doikov et al. (2023) and assume the complexity of obtaining a first-order oracle as $N$ and the complexity of obtaining a second-order oracle as $dN$. In this paper, we show that the computation cost can be reduced by reusing Hessian across iterations. Our methods take the overall computational complexity of $\tilde{\mathcal{O}}((N + d^2)(d + d^{2/3}\epsilon^{-2/3}))$, which improves those of the previous methods by a factor of $d^{1/3}$. Furthermore, we generalize our method to strongly-convex-strongly-concave minimax problems and establish the complexity of $\tilde{\mathcal{O}}((N + d^2)(d + d^{2/3}\kappa^{2/3}))$ when the condition number of the problem is $\kappa$, enjoying a similar speedup upon the state-of-the-art method. Numerical experiments on both real and synthetic datasets also verify the efficiency of our method.

## 1 Introduction

We consider the following minimax optimization problem:

$$\min_{\boldsymbol{x} \in \mathbb{R}^{d_x}} \max_{\boldsymbol{y} \in \mathbb{R}^{d_y}} f(\boldsymbol{x}, \boldsymbol{y}), \tag{1}$$

where we suppose $f(\boldsymbol{x}, \boldsymbol{y})$ is (strongly-)convex in $\boldsymbol{x}$ and (strongly-)concave in $\boldsymbol{y}$. This setting covers many useful applications, including functionally constrained optimization (Xu, 2020), game theory (Von Neumann & Morgenstern, 1947), robust optimization (Ben-Tal et al., 2009), fairness-aware machine learning (Zhang et al., 2018), reinforcement learning (Du et al., 2017; Wang, 2017; Paternain et al., 2022; Wai et al., 2018), decentralized optimization (Kovalev et al., 2021; 2020), AUC maximization (Ying et al., 2016; Hanley & McNeil, 1982; Yuan et al., 2021).

First-order methods are widely studied for this problem. Classical algorithms include ExtraGradient (EG) (Korpelevich, 1976; Nemirovski, 2004), Optimistic Gradient Descent Ascent (OGDA) (Popov, 1980; Mokhtari et al., 2020a;b), Hybrid Proximal Extragradient (HPE) (Monteiro & Svaiter, 2010), and Dual Extrapolation (DE) (Nesterov & Scrimali, 2006; Nesterov, 2007). When the gradient of $f(\cdot, \cdot)$ is $L$-Lipschitz continuous, these methods achieve the rate of $\mathcal{O}(\epsilon^{-1})$ under the convex-concave (C-C) setting and the rate of $\mathcal{O}((L/\mu) \log(\epsilon^{-1}))$ when $f(\cdot, \cdot)$ is $\mu$-strongly convex in $\boldsymbol{x}$ and $\mu$-strongly-concave in $\boldsymbol{y}$ (SC-SC) for $\mu > 0$. They are all optimal in C-C and SC-SC settings due to the lower bounds reported by (Nemirovskij & Yudin, 1983; Zhang et al., 2022a).

---

[*] Equal contributions.

[†] The corresponding author.

Second-order methods usually lead to faster rates than first-order methods when the Hessian of $f(\cdot, \cdot)$ is $\rho$-Lipschitz continuous. A line of works (Nesterov & Scrimali, 2006; Huang et al., 2022) extended the celebrated Cubic Regularized Newton (CRN) (Nesterov & Polyak, 2006) method to minimax problems with local superlinear convergence rates and global convergence guarantee. However, the established global convergence rates of $\mathcal{O}(\epsilon^{-1})$ by Nesterov & Scrimali (2006) and $\mathcal{O}((L\rho/\mu^2)\log(\epsilon^{-1}))$ by Huang et al. (2022) under C-C and SC-SC conditions are no better than the optimal first-order methods. Another line of work generalizes the optimal first-order methods to higher-order methods. Monteiro & Svaiter (2012) proposed the Newton Proximal Extragradient (NPE) method with a global convergence rate of $\mathcal{O}(\epsilon^{-2/3}\log\log(\epsilon^{-1}))$ under the C-C conditions. This result nearly matches the lower bounds (Adil et al., 2022; Lin & Jordan, 2024), except an additional $\mathcal{O}(\log\log(\epsilon^{-1}))$ factor which is caused by the implicit binary search at each iteration. Bullins & Lai (2022); Adil et al. (2022); Huang & Zhang (2022); Lin et al. (2022) provided a simple proof of NPE motivated by the EG analysis and showed that replacing the quadratic regularized Newton step with the cubic regularized Newton (CRN) step in NPE achieves the optimal second-order oracle complexity of $\mathcal{O}(\epsilon^{-2/3})$. Recently, Alves & Svaiter (2023) proposed a search-free NPE method to achieve the optimal second-order oracle complexity with pure quadratic regularized Newton step based on ideas from homotopy. Over the past decade, researchers also proposed various second-order methods, in addition to the NPE framework, that achieve the same convergence rate, such as the second-order extensions of OGDA (Jiang & Mokhtari, 2022; Jiang et al., 2024) (which we refer to as OGDA-2) and DE (Lin & Jordan, 2024) (they name the method Persesus). The results for C-C problems can also be extended to SC-SC problems, where Jiang & Mokhtari (2022) proved the OGDA-2 can converge at the rate of $\mathcal{O}((\rho/\mu)^{2/3} + \log\log(\epsilon^{-1}))$, and Huang & Zhang (2022) proposed the ARE-restart with the rate of $\mathcal{O}((\rho/\mu)^{2/3}\log\log(\epsilon^{-1}))$.

Although the aforementioned second-order methods Adil et al. (2022); Lin & Jordan (2024); Lin et al. (2022); Jiang & Mokhtari (2022); Monteiro & Svaiter (2012) enjoy an improved convergence rate over the first-order methods and have achieved optimal iteration complexities, they require querying one new Hessian at each iteration and solving a matrix inversion problem at each Newton step, which leads to a $\mathcal{O}(d^3)$ computational cost per iteration. This becomes the main bottleneck that limits the applicability of second-order methods. Liu & Luo (2022a) proposed quasi-Newton methods for saddle point problems that access one Hessian-vector product instead of the exact Hessian for each iteration. The iteration complexity is $\mathcal{O}(d^2)$ for quasi-Newton methods. However, their methods do not have a global convergence guarantee under general (S)C-(S)C conditions. Jiang et al. (2023) proposed an online-learning guided Quasi-Newton Proximal Extragradient (QNPE) algorithm, but their method relies on more complicated subroutines than classical Newton methods. Although the oracle complexity of QNPE is strictly better than the optimal first-order method EG, their method is worse in terms of total computational complexity.

In this paper, we propose a computation-efficient second-order method, which we call LEN (Lazy Extra Newton method). In contrast to all existing second-order methods or quasi-Newton methods for minimax optimization problems that always access new second-order information for the coming iteration, LEN reuses the second-order information from past iterations. Specifically, LEN solves a cubic regularized sub-problem using the Hessian from the snapshot point that is updated every $m$ iteration, then conducts an extra-gradient step by the gradient from the current iteration. We provide a rigorous theoretical analysis of LEN to show it maintains fast global convergence rates and improves the (near)-optimal second-order methods (Monteiro & Svaiter, 2012) in terms of the overall computational complexity. We summarize our contributions as follows (also see Table 1).

- When the object function $f(\cdot, \cdot)$ is convex in $\boldsymbol{x}$ and concave in $\boldsymbol{y}$, we propose LEN and prove that it finds an $\epsilon$-saddle point in $\mathcal{O}(m^{2/3}\epsilon^{-2/3})$ iterations. Under Assumption 3.4, where the complexity of calculating $\boldsymbol{F}(\boldsymbol{z})$ is $N$ and the complexity of calculating $\nabla\boldsymbol{F}(\boldsymbol{z})$ is $dN$, the optimal choice is $m = \Theta(d)$. In this case, LEN only requires a computational complexity of $\tilde{\mathcal{O}}((N+d^2)(d+d^{2/3}\epsilon^{-2/3}))$, which is strictly better than $\mathcal{O}((N+d^2)d\epsilon^{-2/3})$ for the existing optimal second-order methods by a factor of $d^{1/3}$.

- When the object function $f(\cdot, \cdot)$ is $\mu$-strongly-convex in $\boldsymbol{x}$ and $\mu$-strongly-concave in $\boldsymbol{y}$, we apply the restart strategy on LEN and propose LEN-restart. We prove the algorithm can find an $\epsilon$-root with $\tilde{\mathcal{O}}((N+d^2)(d+d^{2/3}(\rho/\mu)^{2/3}))$ computational complexity, where $\rho$ means the Hessian of $f(\cdot, \cdot)$ is $\rho$ Lipschitz-continuous. Our result is strictly better than the $\tilde{\mathcal{O}}((N+d^2)d(\rho/\mu)^{2/3})$ in prior works.

Table 1: We compare the required computational complexity to achieve an $\epsilon$-saddle point of the proposed LEN with the optimal choice $m = \Theta(d)$ and other existing algorithms on both convex-concave (C-C) and strongly-convex-strongly-concave (SC-SC) problems. Here, $d = d_x + d_y$ is the dimension of the problem. We assume the gradient is $L$-Lipschitz continuous for EG and the Hessian is $\rho$-Lipschitz continuous for others. We count each gradient oracle call with $N$ computational complexity, and each Hessian oracle with $dN$ computational complexity.

| Setup | Method | Computational Cost |
|-------|--------|--------------------|
| C-C | EG (Korpelevich, 1976) | $\mathcal{O}((N+d)\epsilon^{-1})$ |
| | NPE (Monteiro & Svaiter, 2012) | $\tilde{\mathcal{O}}((N+d^2)d\epsilon^{-2/3})$ |
| | search-free NPE (Alves & Svaiter, 2023) | $\mathcal{O}((N+d^2)d\epsilon^{-2/3})$ |
| | OGDA-2 (Jiang & Mokhtari, 2022) | $\mathcal{O}((N+d^2)d\epsilon^{-2/3})$ |
| | LEN (Theorem 4.3) | $\tilde{\mathcal{O}}((N+d^2)(d+d^{2/3}\epsilon^{-2/3}))$ |
| SC-SC | EG (Korpelevich, 1976) | $\tilde{\mathcal{O}}((N+d)(L/\mu))$ |
| | OGDA-2 (Jiang & Mokhtari, 2022) | $\mathcal{O}((N+d^2)d(\rho/\mu)^{2/3})$ |
| | ARE-restart (Huang & Zhang, 2022) | $\tilde{\mathcal{O}}((N+d^2)d(\rho/\mu))^{2/3})$ |
| | Perseus-restart (Lin & Jordan, 2024) | $\tilde{\mathcal{O}}((N+d^2)d(\rho/\mu)^{2/3})$ |
| | LEN-restart (Corollary 4.1) | $\tilde{\mathcal{O}}((N+d^2)(d+d^{2/3}(\rho/\mu)^{2/3}))$ |

**Notations.** Throughout this paper, $\log$ is base 2 and $\log_+(\,\cdot\,) := 1 + \log(\,\cdot\,)$. We use $\|\cdot\|$ to denote the spectral norm and the Euclidean norm of matrices and vectors, respectively. We denote $\pi(t) = t - (t \mod m)$ where $m \in \mathbb{N}_+$.

## 2 RELATED WORKS AND TECHNICAL CHALLENGES

**Lazy Hessian in minimization problems.** The idea of reusing Hessian was initially presented by Shamanskii (1967) and later incorporated into the Levenberg-Marquardt method, the Damped Newton method, and the proximal Newton method (Fan, 2013; Lampariello & Sciandrone, 2001; Wang et al., 2006; Adler et al., 2020). However, the explicit advantage of lazy Hessian update over ordinary Newton (-type) update was not discovered until the recent work of (Doikov et al., 2023; Chayti et al., 2023). They applied the following lazy Hessian update on cubic regularized Newton (CRN) methods (Nesterov & Polyak, 2006):

$$\boldsymbol{z}_{t+1} = \arg\min_{\boldsymbol{z} \in \mathbb{R}^d} \left\{ \langle \boldsymbol{F}(\boldsymbol{z}_t), \boldsymbol{z} - \boldsymbol{z}_t \rangle + \frac{1}{2} \langle \nabla \boldsymbol{F}(\boldsymbol{z}_{\pi(t)})(\boldsymbol{z} - \boldsymbol{z}_t), \boldsymbol{z} - \boldsymbol{z}_t \rangle + \frac{M}{6} \|\boldsymbol{z} - \boldsymbol{z}_t\|^3 \right\}, \quad (2)$$

where $M \geq 0$ and $\boldsymbol{F} : \mathbb{R}^d \to \mathbb{R}^d$ is the gradient field of a convex function. They establish the convergence rates of $\mathcal{O}(\sqrt{m}\epsilon^{-3/2})$ for nonconvex optimization (Doikov et al., 2023), and $\mathcal{O}(\sqrt{m}\epsilon^{-1/2})$ for convex optimization (Chayti et al., 2023) respectively. Such rates lead to the total computational cost of $\tilde{\mathcal{O}}((N+d^2)(d+\sqrt{d}\epsilon^{-3/2}))$ and $\tilde{\mathcal{O}}((N+d^2)(d+\sqrt{d}\epsilon^{-1/2}))$ by setting $m = \Theta(d)$, which strictly improve the result by classical CRN methods by a factor of $\sqrt{d}$ in both setups.

We have also observed that the idea of the "lazy Hessian" is widely used in practical second-order methods. KFAC (Martens & Grosse, 2015; Grosse & Martens, 2016) approximates the Fisher information matrix and uses an exponential moving average (EMA) to update the estimate of the Fisher information matrix, which can be viewed as a soft version of lazy update. Sophia (Liu et al., 2024) estimates a diagonal Hessian matrix as a pre-conditioner, which is updated in a lazy manner to reduce the complexity. C2EDEN (Liu et al., 2023) atomizes the communication of local Hessian in several consecutive iterations, which also benefits from the idea of lazy updates.

**Challenge of using lazy Hessian updates in minimax problems.** In comparison to previous work on lazy Hessian, our LEN and LEN-restart methods demonstrate the advantage of using lazy Hessian

for a broader class of optimization problems, the *minimax* problems. Our analysis differs from the ones in Doikov et al. (2023); Chayti et al. (2023). Their methods only take a lazy CRN update (2) at each iteration, which makes it easy to bound the error of lazy Hessian updates using Assumption 3.1 and the triangle inequality in the following way:

$$\|\nabla \boldsymbol{F}(\boldsymbol{z}_t) - \nabla \boldsymbol{F}(\boldsymbol{z}_{\pi(t)})\| \le \rho \|\boldsymbol{z}_{\pi(t)} - \boldsymbol{z}_t\| \le \rho \sum_{i=\pi(t)}^{t-1} \|\boldsymbol{z}_i - \boldsymbol{z}_{i+1}\|.$$

Our method, on the other hand, not only takes a lazy (regularized) Newton update but also requires an extra gradient step (Line 4 in Algorithm 1). Thus, the progress of one Newton update $\{\|\boldsymbol{z}_{i+1/2} - \boldsymbol{z}_i\|\}_{i=\pi(t)}^{t}$ cannot directly bound the error term $\|\boldsymbol{z}_t - \boldsymbol{z}_{\pi(t)}\|$ introduced by the lazy Hessian update. Moreover, for minimax problems the matrix $\nabla \boldsymbol{F}(\boldsymbol{z}_{\pi(t)})$ is no longer symmetric, which leads to different analysis and implementation of sub-problem solving (Section 4.3). We refer the readers to Section 4 for more detailed discussions.

## 3 PRELIMINARIES

In this section, we introduce the notation and basic assumptions used in our work. We start with several standard definitions for Problem (1).

**Definition 3.1.** *We call a function $f(\boldsymbol{x}, \boldsymbol{y}) : \mathbb{R}^{d_x} \times \mathbb{R}^{d_y} \to \mathbb{R}$ has $\rho$-Lipschitz Hessians if we have*

$$\|\nabla^2 f(\boldsymbol{x}, \boldsymbol{y}) - \nabla^2 f(\boldsymbol{x}', \boldsymbol{y}')\| \le \rho \left\| \begin{bmatrix} \boldsymbol{x} - \boldsymbol{x}' \\ \boldsymbol{y} - \boldsymbol{y}' \end{bmatrix} \right\|, \quad \forall (\boldsymbol{x}, \boldsymbol{y}), (\boldsymbol{x}', \boldsymbol{y}') \in \mathbb{R}^{d_x} \times \mathbb{R}^{d_y}.$$

**Definition 3.2.** *A differentiable function $f(\cdot, \cdot)$ is $\mu$-strongly-convex-$\mu$-strongly-concave for some $\mu > 0$ if*

$$f(\boldsymbol{x}', \boldsymbol{y}) \ge f(\boldsymbol{x}, \boldsymbol{y}) + (\boldsymbol{x}' - \boldsymbol{x})^\top \nabla_x f(\boldsymbol{x}, \boldsymbol{y}) + \frac{\mu}{2} \|\boldsymbol{x} - \boldsymbol{x}'\|^2, \quad \forall \boldsymbol{x}', \boldsymbol{x} \in \mathbb{R}^{d_x}, \boldsymbol{y} \in \mathbb{R}^{d_y};$$

$$f(\boldsymbol{x}, \boldsymbol{y}') \le f(\boldsymbol{x}, \boldsymbol{y}) + (\boldsymbol{y}' - \boldsymbol{y})^\top \nabla_y f(\boldsymbol{x}, \boldsymbol{y}) - \frac{\mu}{2} \|\boldsymbol{y} - \boldsymbol{y}'\|^2, \quad \forall \boldsymbol{y}', \boldsymbol{y} \in \mathbb{R}^{d_y}, \boldsymbol{x} \in \mathbb{R}^{d_x}.$$

*We say $f$ is convex-concave if $\mu = 0$.*

We are interested in finding a saddle point of Problem (1), formally defined as follows.

**Definition 3.3.** *We call a point $(\boldsymbol{x}^*, \boldsymbol{y}^*) \in \mathbb{R}^{d_x} \times \mathbb{R}^{d_y}$ a saddle point of a function $f(\cdot, \cdot)$ if we have*

$$f(\boldsymbol{x}^*, \boldsymbol{y}) \le f(\boldsymbol{x}^*, \boldsymbol{y}^*) \le f(\boldsymbol{x}, \boldsymbol{y}^*), \quad \forall \boldsymbol{x} \in \mathbb{R}^{d_x}, \ \boldsymbol{y} \in \mathbb{R}^{d_y}.$$

Next, we introduce all the assumptions made in this work. In this paper, we focus on Problem (1) that satisfies the following assumptions.

**Assumption 3.1.** *We assume the function $f(\cdot, \cdot)$ is twice continuously differentiable, has $\rho$-Lipschitz continuous Hessians, and has at least one saddle point $(\boldsymbol{x}^*, \boldsymbol{y}^*)$.*

We will study convex-concave problems and strongly-convex-strongly-concave problems.

**Assumption 3.2** (C-C setting). *We assume the function $f(\cdot, \cdot)$ is convex in $\boldsymbol{x}$ and concave in $\boldsymbol{y}$.*

**Assumption 3.3** (SC-SC setting). *We assume the function $f(\cdot, \cdot)$ is $\mu$-strongly-convex-$\mu$-strongly-concave. We denote the condition number as $\kappa := \rho/\mu$.*

We let $d := d_x + d_y$ and denote the aggregated variable $\boldsymbol{z} := (\boldsymbol{x}, \boldsymbol{y}) \in \mathbb{R}^d$. We also denote the GDA field of $f$ and its Jacobian as

$$\boldsymbol{F}(\boldsymbol{z}) := \begin{bmatrix} \nabla_x f(\boldsymbol{x}, \boldsymbol{y}) \\ -\nabla_y f(\boldsymbol{x}, \boldsymbol{y}) \end{bmatrix}, \quad \nabla \boldsymbol{F}(\boldsymbol{z}) := \begin{bmatrix} \nabla_{xx}^2 f(\boldsymbol{x}, \boldsymbol{y}) & \nabla_{xy}^2 f(\boldsymbol{x}, \boldsymbol{y}) \\ -\nabla_{yx}^2 f(\boldsymbol{x}, \boldsymbol{y}) & -\nabla_{yy}^2 f(\boldsymbol{x}, \boldsymbol{y}) \end{bmatrix}. \tag{3}$$

The GDA field of $f(\cdot, \cdot)$ has the following properties.

**Lemma 3.1** (Lemma 2.7 Lin et al. (2022)). *Under Assumptions 3.1 and 3.2, we have*

1. *$\boldsymbol{F}$ is monotone, i.e. $\langle \boldsymbol{F}(\boldsymbol{z}) - \boldsymbol{F}(\boldsymbol{z}'), \boldsymbol{z} - \boldsymbol{z}' \rangle \geq 0$, $\forall \boldsymbol{z}, \boldsymbol{z}' \in \mathbb{R}^d$.*

2. *$\nabla \boldsymbol{F}$ is $\rho$-Lipschitz continuous, i.e. $\|\nabla \boldsymbol{F}(\boldsymbol{z}) - \nabla \boldsymbol{F}(\boldsymbol{z}')\| \leq \rho \|\boldsymbol{z} - \boldsymbol{z}'\|$, $\forall \boldsymbol{z}, \boldsymbol{z}' \in \mathbb{R}^d$.*

3. *$\boldsymbol{F}(\boldsymbol{z}^*) = 0$ if and only if $\boldsymbol{z}^* = (\boldsymbol{x}^*, \boldsymbol{y}^*)$ is a saddle point of function $f(\cdot, \cdot)$.*

*Furthermore, if Assumption 3.3 holds, we have $\boldsymbol{F}(\cdot)$ is $\mu$-strongly-monotone, i.e.*

$$\langle \boldsymbol{F}(\boldsymbol{z}) - \boldsymbol{F}(\boldsymbol{z}'), \boldsymbol{z} - \boldsymbol{z}' \rangle \geq \mu \|\boldsymbol{z} - \boldsymbol{z}'\|^2, \ \forall \boldsymbol{z}, \boldsymbol{z}' \in \mathbb{R}^d.$$

For the C-C case, the commonly used optimality criterion is the following restricted gap.

**Definition 3.4** (Nesterov (2007)). *Let $\mathbb{B}_\beta(\boldsymbol{w})$ be the ball centered at $\boldsymbol{w}$ with radius $\beta$. Let $(\boldsymbol{x}^*, \boldsymbol{y}^*)$ be a saddle point of function $f$. For a given point $(\hat{\boldsymbol{x}}, \hat{\boldsymbol{y}})$, we let $\beta$ sufficiently large such that it holds*

$$\max \left\{ \|\hat{\boldsymbol{x}} - \boldsymbol{x}^*\|, \ \|\hat{\boldsymbol{y}} - \boldsymbol{y}^*\| \right\} \leq \beta,$$

*we define the restricted gap function as*

$$\mathrm{Gap}(\hat{\boldsymbol{x}}, \hat{\boldsymbol{y}}; \beta) := \max_{\boldsymbol{y} \in \mathbb{B}_\beta(\boldsymbol{y}^*)} f(\hat{\boldsymbol{x}}, \boldsymbol{y}) - \min_{\boldsymbol{x} \in \mathbb{B}_\beta(\boldsymbol{x}^*)} f(\boldsymbol{x}, \hat{\boldsymbol{y}}),$$

*We call $(\hat{\boldsymbol{x}}, \hat{\boldsymbol{y}})$ an $\epsilon$-saddle point if $\mathrm{Gap}(\hat{\boldsymbol{x}}, \hat{\boldsymbol{y}}; \beta) \leq \epsilon$ and $\beta = \Omega(\max\{\|\boldsymbol{x}_0 - \boldsymbol{x}^*\|, \|\boldsymbol{y}_0 - \boldsymbol{y}^*\|\})$.*

For the SC-SC case, we use the following stronger notion.

**Definition 3.5.** *Suppose that Assumption 3.3 holds. Let $\boldsymbol{z}^* = (\boldsymbol{x}^*, \boldsymbol{y}^*)$ be the unique saddle point of function $f$. We call $\hat{\boldsymbol{z}} = (\hat{\boldsymbol{x}}, \hat{\boldsymbol{y}})$ an $\epsilon$-root if $\|\hat{\boldsymbol{z}} - \boldsymbol{z}^*\| \leq \epsilon$.*

Most previous works only consider the complexity metric as the number of oracle calls, where an oracle takes a point $\boldsymbol{z} \in \mathbb{R}^d$ as the input and returns a tuple $(\boldsymbol{F}(\boldsymbol{z}), \nabla \boldsymbol{F}(\boldsymbol{z}))$ as the output. The existing algorithms (Monteiro & Svaiter, 2012; Bullins & Lai, 2022; Adil et al., 2022; Lin et al., 2022) have achieved optimal complexity regarding the number of oracle calls. In this work, we focus on the computational complexity of the oracle. More specifically, we distinguish between the different computational complexities of calculating the Hessian matrix $\nabla \boldsymbol{F}(\boldsymbol{z})$ and the gradient $\boldsymbol{F}(\boldsymbol{z})$. Formally, we make the following assumption as Doikov et al. (2023).

**Assumption 3.4.** *We count the computational complexity of computing $\boldsymbol{F}(\cdot)$ as $N$ and the computational complexity of $\nabla \boldsymbol{F}(\cdot)$ as $Nd$.*

**Remark 3.1.** *Assumption 3.4 supposes the cost of computing $\nabla \boldsymbol{F}(\cdot)$ is $d$ times that of computing $\boldsymbol{F}(\cdot)$. It holds in many practical scenarios as one Hessian oracle can be computed via $d$ Hessian-vector products on standard basis vectors $\boldsymbol{e}_1, \cdots, \boldsymbol{e}_d$, and one Hessian-vector product oracle is typically as expensive as one gradient oracle (Wright, 2006):*

1. *When the computational graph of $f$ is obtainable, both $\boldsymbol{F}(\boldsymbol{z})$ and $\nabla \boldsymbol{F}(\boldsymbol{z})\boldsymbol{v}$ can be computed using automatic differentiation with the same cost for any $\boldsymbol{z}, \boldsymbol{v} \in \mathbb{R}^d$.*

2. *When $f$ is a black box function, we can estimate the Hessian-vector $\nabla \boldsymbol{F}(\boldsymbol{z})\boldsymbol{v}$ via the finite-difference $\boldsymbol{u}_\delta(\boldsymbol{z}; \boldsymbol{v}) = \frac{1}{\delta}(\boldsymbol{F}(\boldsymbol{z} + \delta \boldsymbol{v}) - \boldsymbol{F}(\boldsymbol{z} - \delta \boldsymbol{v}))$ and we have $\lim_{\delta \to 0} \boldsymbol{u}_\delta(\boldsymbol{z}; \boldsymbol{v}) = \nabla \boldsymbol{F}(\boldsymbol{z})\boldsymbol{v}$ under mild conditions on $\boldsymbol{F}(\cdot)$.*

## 4 ALGORITHMS AND CONVERGENCE ANALYSIS

In this section, we present novel second-order methods for solving minimax optimization problems (1). We present LEN and its convergence analysis for convex-concave minimax problems in Section 4.1. We generalize LEN for strongly-convex-strongly-concave minimax problems by presenting LEN-restart in Section 4.2. We discuss the details of solving minimax cubic-regularized sub-problem, present detailed implementation of LEN, and give the total computational complexity of proposed methods in Section 4.3.

### 4.1 THE LEN ALGORITHM FOR CONVEX-CONCAVE PROBLEMS

We propose LEN for convex-concave problems in Algorithm 1. Our method builds on the optimal Newton Proximal Extragradient (NPE) method (Monteiro & Svaiter, 2012; Bullins & Lai, 2022;

---

**Algorithm 1** LEN($z_0, T, m, M$)

---
1: **for** $t = 0, \cdots, T - 1$ **do**
2:     Compute lazy cubic step, *i.e.* find $z_{t+1/2}$ that satisfies

$$F(z_t) = (\nabla F(z_{\pi(t)}) + M\|z_t - z_{t+1/2}\|I_d)(z_t - z_{t+1/2}).$$

3:     Compute $\gamma_t = M\|z_t - z_{t+1/2}\|$.
4:     Compute extra-gradient step $z_{t+1} = z_t - \gamma_t^{-1}F(z_{t+1/2})$.
5: **end for**
6: **return** $\bar{z}_T = \frac{1}{\sum_{t=0}^{T-1}\gamma_t^{-1}} \sum_{t=0}^{T-1} \gamma_t^{-1} z_{t+1/2}$.

---

Adil et al., 2022; Lin et al., 2022). The only change is that we reuse the Hessian from previous iterates, as colored in blue. Each iteration of LEN contains the following two steps:

$$\begin{cases} F(z_t) + \nabla F(z_{\pi(t)})(z_{t+1/2} - z_t) + M\|z_{t+1/2} - z_t\|(z_{t+1/2} - z_t) = 0, & \text{(Implicit Step)} \\ z_{t+1} = z_t - \dfrac{F(z_{t+1/2})}{M\|z_{t+1/2} - z_t\|}. & \text{(Explicit Step)} \end{cases}$$

$$(4)$$

The first step (implicit step) solves a cubic regularized sub-problem based on the $\nabla F(z_{\pi(t)})$ computed at the latest snapshot point and $F(z_t)$ at the current iteration point. This step is often viewed as an oracle (Bullins & Lai, 2022; Adil et al., 2022; Lin et al., 2022) as there exists efficient solvers, which will also be discussed in Section 4.3. The second one (explicit step) conducts an extra gradient step based on $F(z_{t+1/2})$.

Reusing the Hessian in the implicit step makes each iteration much cheaper, but would cause additional errors compared to previous methods (Monteiro & Svaiter, 2012; Huang & Zhang, 2022; Adil et al., 2022; Lin et al., 2022). The error resulting from the lazy Hessian updates is formally characterized by the following theorem.

**Lemma 4.1.** *Suppose that Assumption 3.1 and 3.2 hold. For any $z \in \mathbb{R}^d$, Algorithm 1 ensures*

$$\gamma_t^{-1}\langle F(z_{t+1/2}), z_{t+1/2} - z \rangle \leq \frac{1}{2}\|z_t - z\|^2 - \frac{1}{2}\|z_{t+1} - z\|^2 - \frac{1}{2}\|z_{t+1/2} - z_{t+1}\|^2$$
$$- \frac{1}{2}\|z_t - z_{t+1/2}\|^2 + \frac{\rho^2}{2M^2}\|z_t - z_{t+1/2}\|^2 + \underbrace{\frac{2\rho^2}{M^2}\|z_{\pi(t)} - z_t\|^2}_{(*)}.$$

Above, (*) is the error from lazy Hessian updates. Note that (*) vanishes when the current Hessian is used. For lazy Hessian updates, the error would accumulate in the epoch.

The key step in our analysis shows that we can use the negative terms in the right-hand side of the inequality in Lemma 4.1 to bound the accumulated error by choosing $M$ sufficiently large, with the help of the following technical lemma.

**Lemma 4.2.** *For any sequence of positive numbers $\{r_t\}_{t\geq0}$, it holds for any $m \geq 2$ that* $\sum_{t=1}^{m-1}\left(\sum_{i=0}^{t-1} r_i\right)^2 \leq \frac{m^2}{2}\sum_{t=0}^{m-1} r_t^2.$

When $m = 1$, the algorithm reduces to the NPE algorithm (Monteiro & Svaiter, 2012; Bullins & Lai, 2022; Adil et al., 2022; Lin et al., 2022) without using lazy Hessian updates. When $m \geq 2$, we use Lemma 4.2 to upper bound the error that arises from lazy Hessian updates. Finally, we prove the following guarantee for our proposed algorithm.

**Theorem 4.1** (C-C setting). *Suppose that Assumption 3.1 and 3.2 hold. Let $z^* = (x^*, y^*)$ be a saddle point and $\beta = \|z_0 - z^*\|$. Set $M \geq 3\rho m$. The sequence of iterates generated by Algorithm 1 is bounded $z_t \in \mathbb{B}_\beta(z^*)$, $z_{t+1/2} \in \mathbb{B}_{3\beta}(z^*)$, $\forall t = 0, \cdots, T - 1$, and satisfies the following ergodic convergence:*

$$\text{Gap}(\bar{x}_T, \bar{y}_T; 3\beta) \leq \frac{32M\|z_0 - z^*\|^3}{T^{3/2}}.$$

*Let $M = 3\rho m$. Algorithm 1 finds an $\epsilon$-saddle point within $\mathcal{O}(m^{2/3}\epsilon^{-2/3})$ iterations.*

---

**Algorithm 2** LEN-restart($\boldsymbol{z}_0, T, m, M, S$)

1: $\boldsymbol{z}^{(0)} = \boldsymbol{z}_0$
2: **for** $s = 0, \cdots, S-1$
3: $\quad \boldsymbol{z}^{(s+1)} = \text{LEN}(\boldsymbol{z}^{(s)}, T, m, M)$
  **end for**

---

**Discussion on the computational complexity of the oracles.** Theorem 4.1 indicates that LEN requires $\mathcal{O}(m^{2/3}\epsilon^{-2/3})$ calls to $\boldsymbol{F}(\cdot)$ and $\mathcal{O}(m^{2/3}\epsilon^{-2/3}/m + 1)$ calls to $\nabla\boldsymbol{F}(\cdot)$ to find the $\epsilon$-saddle point. Under Assumption 3.4, the computational cost to call the oracles $\boldsymbol{F}(\cdot)$ and $\nabla\boldsymbol{F}(\cdot)$ is

$$\text{Oracle Computational Cost} = \mathcal{O}\left(N \cdot m^{2/3}\epsilon^{-2/3} + (Nd)\cdot\left(\epsilon^{-2/3}m^{-1/3} + 1\right)\right). \quad (5)$$

Taking $m = \Theta(d)$ minimizes (5) to $\mathcal{O}(Nd + Nd^{2/3}\epsilon^{-2/3})$. Compared to state-of-the-art second-order methods (Monteiro & Svaiter, 2012; Bullins & Lai, 2022; Adil et al., 2022; Lin et al., 2022), whose computational cost in terms of the oracles is $\mathcal{O}(Nd\epsilon^{-2/3})$ since they require to query $\nabla\boldsymbol{F}(\cdot)$ at each iteration, our methods significantly improve their results by a factor of $d^{1/3}$.

It is worth noticing that the computational cost of an algorithm includes both the computational cost of calling the oracles, which we have discussed above, and the computational cost of performing the updates (*i.e.* solving auxiliary problems) after accessing the required oracles. We will give an efficient implementation of LEN and analyze the total computational cost later in Section 4.3.

## 4.2 THE LEN-RESTART ALGORITHM FOR STRONGLY-CONVEX-STRONGLY-CONCAVE PROBLEMS

We generalize LEN to solve the strongly-convex-strongly-concave minimax optimization by incorporating the restart strategy introduced by Huang & Zhang (2022); Lin & Jordan (2024). We propose the LEN-restart in Algorithm 2, which works in epochs. Each epoch of LEN-restart invokes LEN (Algorithm 1), which gets $\boldsymbol{z}^{(s)}$ as inputs and outputs $\boldsymbol{z}^{(s+1)}$.

The following theorem shows that the sequence $\{\boldsymbol{z}^{(s)}\}$ enjoys a superlinear convergence in epochs. Furthermore, the required number of iterations in each epoch to achieve such a superlinear rate is only a constant.

**Theorem 4.2** (SC-SC setting). *Suppose that Assumptions 3.1 and 3.3 hold. Let $\boldsymbol{z}^* = (\boldsymbol{x}^*, \boldsymbol{y}^*)$ be the unique saddle point. Set $M = 3\rho m$ as Theorem 4.1 and $T = \left(\frac{2M\|\boldsymbol{z}_0 - \boldsymbol{z}^*\|}{\mu}\right)^{2/3}$. Then the sequence of iterates generated by Algorithm 2 converge to $\boldsymbol{z}^*$ superlinearly as $\|\boldsymbol{z}^{(s)} - \boldsymbol{z}^*\|^2 \leq \left(\frac{1}{2}\right)^{(3/2)^s}\|\boldsymbol{z}_0 - \boldsymbol{z}^*\|^2$. In other words, Algorithm 2 finds a point $\boldsymbol{z}^{(s)}$ such that $\|\boldsymbol{z}^{(s)} - \boldsymbol{z}^*\| \leq \epsilon$ within $S = \log_{3/2}\log_2(1/\epsilon)$ epochs. The total number of inner loop iterations is given by*

$$TS = \mathcal{O}\left(m^{2/3}\kappa^{2/3}\log\log(1/\epsilon)\right).$$

*Under Assumption 3.4, Algorithm 2 with $m = \Theta(d)$ takes the computational complexity of $\mathcal{O}((Nd + Nd^{2/3}\kappa^{2/3})\log\log(1/\epsilon))$ to call the oracles $\boldsymbol{F}(\cdot)$ and $\nabla\boldsymbol{F}(\cdot)$.*

## 4.3 IMPLEMENTATION DETAILS AND COMPUTATIONAL COMPLEXITY ANALYSIS

We provide details of implementing the cubic regularized Newton oracle (Implicit Step, (4)). Inspired by Monteiro & Svaiter (2012); Bullins & Lai (2022); Adil et al. (2022); Lin et al. (2022), we transform the sub-problem into a root-finding problem for a univariate function.

**Lemma 4.3** (Section 4.3 Lin et al. (2022)). *Suppose Assumption 3.1 and 3.2 hold for function $f : \mathbb{R}^{d_x} \times \mathbb{R}^{d_y} \to \mathbb{R}$ and let $\boldsymbol{F}$ be its GDA field. Define $\gamma_t = M\|\boldsymbol{z}_{t+1/2} - \boldsymbol{z}_t\|$. The cubic regularized Newton oracle (Implicit Step, (4)) can be rewritten as:*

$$\boldsymbol{z}_{t+1/2} = \boldsymbol{z}_t - (\nabla\boldsymbol{F}(\boldsymbol{z}_{\pi(t)}) + \gamma_t\boldsymbol{I}_d)^{-1}\boldsymbol{F}(\boldsymbol{z}_t),$$

*which can be implemented by finding the root of the following univariate function:*

$$\phi(\gamma) := M\left\|(\nabla\boldsymbol{F}(\boldsymbol{z}_{\pi(t)}) + \gamma\boldsymbol{I}_d)^{-1}\boldsymbol{F}(\boldsymbol{z}_t)\right\| - \gamma. \quad (6)$$

*Furthermore, the function $\phi(\gamma)$ is strictly decreasing when $\lambda > 0$.*

---

**Algorithm 3** Implementation of LEN $(\boldsymbol{z}_0, T, m, M)$

---

1: **for** $t = 0, \cdots, T - 1$ **do**
2:      **if** $t \mod m = 0$ **do**
3:          Compute the Schur decomposition such that $\nabla \boldsymbol{F}(\boldsymbol{z}_t) = \boldsymbol{Q}\boldsymbol{U}\boldsymbol{Q}^{-1}$.
4:      **end if**
5:      Let $\phi(\,\cdot\,)$ defined as 6 and compute $\gamma_t$ as its root by a binary search.
6:      Compute lazy cubic step $\boldsymbol{z}_{t+1/2} = \boldsymbol{Q}(\boldsymbol{U} + \gamma_t \boldsymbol{I}_d)^{-1}\boldsymbol{Q}^{-1}\boldsymbol{F}(\boldsymbol{z}_t)$.
7:      Compute extra-gradient step $\boldsymbol{z}_{t+1} = \boldsymbol{z}_t - \gamma_t^{-1}\boldsymbol{F}(\boldsymbol{z}_{t+1/2})$.
8: **end for**
9: **return** $\bar{\boldsymbol{z}}_T = \frac{1}{\sum_{t=0}^{T-1}\gamma_t^{-1}} \sum_{t=0}^{T-1} \gamma_t^{-1}\boldsymbol{z}_{t+1/2}$.

---

From the above lemma, to implement the cubic regularized Newton oracle, it suffices to find the root of a strictly decreasing function $\phi(\gamma)$, which can be solved within $\tilde{\mathcal{O}}(1)$ iteration. The main operation is to solve the following linear system:

$$(\nabla \boldsymbol{F}(\boldsymbol{z}_{\pi(t)}) + \gamma \boldsymbol{I}_d)\boldsymbol{h} = \boldsymbol{F}(\boldsymbol{z}_t). \tag{7}$$

Naively solving this linear system at every iteration still results in an expensive computational complexity of $\mathcal{O}(d^3)$ per iteration.

We present a computationally efficient way to implement LEN by leveraging the Schur factorization at the snapshot point $\nabla \boldsymbol{F}(\boldsymbol{z}_{\pi(t)}) = \boldsymbol{Q}\boldsymbol{U}\boldsymbol{Q}^{-1}$, where $\boldsymbol{Q} \in \mathbb{C}^d$ is a unitary matrix and $\boldsymbol{U} \in \mathbb{C}^d$ is an upper-triangular matrix. Then solving the linear system (7) is equivalent to

$$\boldsymbol{h} = \boldsymbol{Q}(\boldsymbol{U} + \gamma \boldsymbol{I}_d)^{-1}\boldsymbol{Q}^{-1}\boldsymbol{F}(\boldsymbol{z}_t). \tag{8}$$

The final implementable algorithm is presented in Algorithm 3.

Now, we are ready to analyze the total computational complexity of LEN, which can be divided into the following two parts:

$$\text{Computational Cost} = \text{Oracle Computational Cost} + \text{Update Computational Cost},$$

where the first part has been discussed in Section 4.1. Regarding the update computational cost, the Schur decomposition with an computational complexity $\mathcal{O}(d^3)$ is required once every $m$ iterations. After Schur's decomposition has been given at the snapshot point, the dominant part of the update computational complexity is solving the upper-triangular linear system (8) with the back substitution algorithm within the computational complexity $\mathcal{O}(d^2)$. Thus, we have

$$\text{Update Computational Cost} = \tilde{\mathcal{O}}\left(d^2 \cdot m^{2/3}\epsilon^{-2/3} + d^3 \cdot \left(\epsilon^{-2/3}m^{-1/3} + 1\right)\right), \tag{9}$$

and the total computational cost of LEN is

$$\text{Computational Cost} \overset{(5),(9)}{=} \tilde{\mathcal{O}}\left((d^2 + N) \cdot m^{2/3}\epsilon^{-2/3} + (d^3 + Nd) \cdot \left(\epsilon^{-2/3}m^{-1/3} + 1\right)\right). \tag{10}$$

By taking $m = \Theta(d)$, we obtain the best computational complexity in (10) of LEN, which is formally stated in the following theorem.

**Theorem 4.3** (C-C setting). *Under the same setting of Theorem 4.1, Algorithm 3 with $m = \Theta(d)$ finds an $\epsilon$-saddle point with computational complexity $\tilde{\mathcal{O}}((N + d^2)(d + d^{2/3}\epsilon^{-2/3}))$.*

We also present the total computational complexity of LEN-restart for SC-SC setting.

**Corollary 4.1** (SC-SC setting). *Under the same setting as in Theorem 4.2, Algorithm 2 implemented in the same way as Algorithm 3 with $m = \Theta(d)$ finds an $\epsilon$-root with computational complexity $\tilde{\mathcal{O}}((N + d^2)(d + d^{2/3}\kappa^{2/3}))$.*

In both cases, our proposed algorithms improve the total computational cost of the optimal second-order methods (Monteiro & Svaiter, 2012; Bullins & Lai, 2022; Adil et al., 2022; Lin et al., 2022) by a factor of $d^{1/3}$.

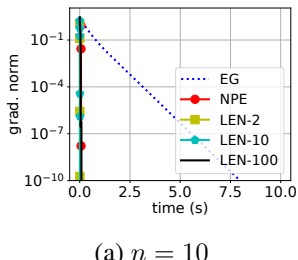 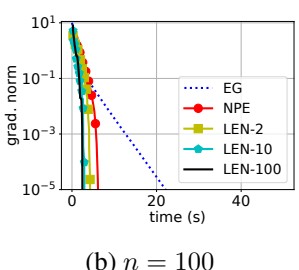 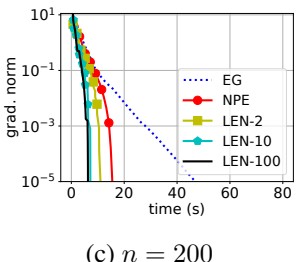

(a) $n = 10$        (b) $n = 100$        (c) $n = 200$

Figure 1: We demonstrate running time *v.s.* gradient norm $\|\boldsymbol{F}(\boldsymbol{z})\|$ for Problem (11) with different sizes: $n \in \{10, 100, 200\}$.

**Remark 4.1.** *In the main text, we assume the use of the classical algorithm for matrix inversion/decomposition, which has a computational complexity of $\mathcal{O}(d^3)$. The fast matrix multiplication proposed by researchers in the field of theoretical computer science only requires a complexity of $d^\omega$, where the best known $\omega$ is currently around 2.371552 (Williams et al., 2024). This also implies faster standard linear algebra operators including Schur decomposition and matrix inversion (Demmel et al., 2007). However, the large hidden constant factors in these fast matrix multiplication algorithms mean that the matrix dimensions necessary for these algorithms to be superior to classical algorithms are much larger than what current computers can effectively handle. Consequently, these algorithms are not always used in practice. We present the computational complexity of using fast matrix operations in Appendix G.*

In Appendix H, we extend our algorithms to allow inexact auxiliary CRN sub-problem solving and analyze the total complexity. Specifically, we design an efficient sub-procedure (Algorithm 5) to solve the CRN sub-problem to desired accuracy in only $\mathcal{O}(\log\log(1/\epsilon))$ number of linear system solving. It tightens the $\mathcal{O}(\log(1/\epsilon))$ iteration complexity in (Bullins & Lai, 2022; Adil et al., 2022). Additionally, (Bullins & Lai, 2022; Adil et al., 2022) assume $\sigma_{\min}(\nabla\boldsymbol{F}(\boldsymbol{z})) \geq \mu$ for some positive constant $\mu$, which makes the problem similar to strongly-convex(-strongly-concave) problems, while our analysis does not require such an assumption.

## 5 NUMERICAL EXPERIMENTS

We conduct our algorithms on a regularized bilinear min-max problem and fairness-aware machine learning tasks. We include EG (Korpelevich, 1976) and NPE (Monteiro & Svaiter, 2012; Bullins & Lai, 2022; Adil et al., 2022; Lin et al., 2022) (which is our algorithm with $m = 1$) as baselines, since they are the optimal first- and second-order methods for convex-concave minimax problems, respectively. We run the programs on an AMD EPYC 7H12 64-Core Processor. [1]

### 5.1 REGULARIZED BILINEAR MIN-MAX PROBLEM

We first conduct numerical experiments on the cubic regularized bilinear min-max problem considered in the literature (Lin et al., 2022; Jiang et al., 2024):

$$\min_{\boldsymbol{x}\in\mathbb{R}^n} \max_{\boldsymbol{y}\in\mathbb{R}^n} f(\boldsymbol{x}, \boldsymbol{y}) = \frac{\rho}{6}\|\boldsymbol{x}\|^3 + \boldsymbol{y}^\top(\boldsymbol{A}\boldsymbol{x} - \boldsymbol{b}). \tag{11}$$

The function $f(\boldsymbol{x}, \boldsymbol{y})$ is convex-concave and has $\rho$-Lipschitz continuous Hessians. The unique saddle point $\boldsymbol{z}^* = (\boldsymbol{x}^*, \boldsymbol{y}^*)$ of $f(\boldsymbol{x}, \boldsymbol{y})$ can be explicitly calculated as $\boldsymbol{x}^* = \boldsymbol{A}^{-1}\boldsymbol{b}$ and $\boldsymbol{y}^* = -\rho\|\boldsymbol{x}^*\|^2(\boldsymbol{A}^\top)^{-1}\boldsymbol{x}^*/2$, so we can compute the distance to $\boldsymbol{z}^*$ to measure the performance of the algorithms. Following Lin et al. (2022), we generate each element in $\boldsymbol{b}$ as independent Rademacher

variables in $\{-1, +1\}$, set $\rho = 1/(20n)$ and the matrix $\boldsymbol{A} = \begin{bmatrix} 1 & -1 & & \\ & \ddots & \ddots & \\ & & 1 & -1 \\ & & & 1 \end{bmatrix}$.

---

[1] The source codes are available at `https://github.com/TrueNobility303/LEN`.

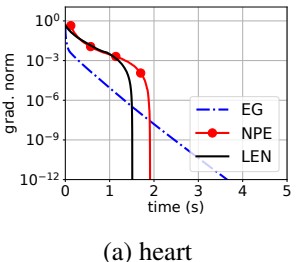 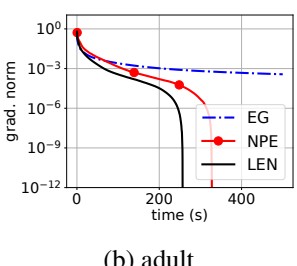 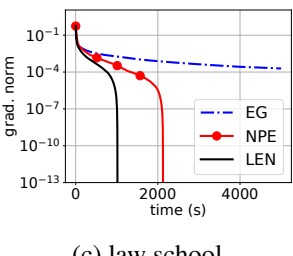

(a) heart          (b) adult          (c) law school

Figure 2: We demonstrate running time *v.s.* gradient norm $\|\boldsymbol{F}(\boldsymbol{z})\|$ for fairness-aware machine learning task (Problem (12)) on datasets "heart", "adult", and "law school".

We compare our methods with the baselines on different sizes of the problem: $n \in \{10, 100, 200\}$. For EG, we tune the stepsize in $\{1, 0.1, 0.01, 0.001\}$. For LEN, we vary $m$ in $\{1, 2, 10, 100\}$. The results of the running time against $\|\boldsymbol{F}(\boldsymbol{z})\|$ are presented in Figure 1.

## 5.2 FAIRNESS-AWARE MACHINE LEARNING

We then examine our algorithm for the task of fairness-aware machine learning. Let $\{\boldsymbol{a}_i, b_i, c_i\}_{i=1}^n$ be the training set, where $\boldsymbol{a}_i \in \mathbb{R}^{d_x}$ denotes the features of the $i$-th sample, $b_i \in \{-1, +1\}$ is the corresponding label, and $c_i \in \{-1, +1\}$ is an additional feature that is required to be protected and debiased. For example, $c_i$ can denote gender. Zhang et al. (2018) proposed to solve the following minimax problem to mitigate unwanted bias of $c_i$ by adversarial learning:

$$\min_{\boldsymbol{x} \in \mathbb{R}^{d_x}} \max_{y \in \mathbb{R}} \frac{1}{n} \sum_{i=1}^n \ell(b_i \boldsymbol{a}_i^\top \boldsymbol{x}) - \beta \ell(c_i y \boldsymbol{a}_i^\top \boldsymbol{x}) + \lambda \|\boldsymbol{x}\|^2 - \gamma y^2, \tag{12}$$

where $\ell$ is the logit function such that $\ell(t) = \log(1 + \exp(-t))$. We set $\lambda = \gamma = 10^{-4}$ and $\beta = 0.5$. We conduct the experiments on datasets "heart" ($n = 270, d_x = 13$) (Chang & Lin, 2011), "adult" ($n = 32,561, d_x = 123$) (Chang & Lin, 2011) and "law school" ($n = 20,798, d_x = 380$) (Le Quy et al., 2022; Liu et al., 2022). For all the datasets, we choose "gender" as the protected feature. For EG, we tune the stepsize in $\{0.1, 0.01, 0.001\}$. For second-order methods (NPE and LEN), as we do not know the value of $\rho$ in advance, we view it as a hyperparameter and tune it in $\{1, 10, 100\}$. We set $m = 10$ for LEN and we find that this simple choice performs well in all the datasets we test. We show the results of the running time against the gradient norm $\|\boldsymbol{F}(\boldsymbol{z})\|$ in Figure 2.

## 6 CONCLUSION AND FUTURE WORKS

In this paper, we propose LEN and LEN-restart for C-C and SC-SC minimax problems, respectively. Using lazy Hessian updates, our methods improve the computational complexity of the current best-known second-order methods by a factor of $d^{1/3}$. Numerical experiments on both real and synthetic datasets also verify the efficiency of our method.

For future work, it will be interesting to extend our idea to adaptive second-order methods (Wang et al., 2024a; Doikov et al., 2024; Carmon et al., 2022; Antonakopoulos et al., 2022; Liu & Luo, 2022b) and stochastic problems with sub-sampled Newton methods (Lin et al., 2022; Chayti et al., 2023; Zhou et al., 2019; Tripuraneni et al., 2018; Wang et al., 2019). Besides, our methods only focus on the convex-concave case, it is also possible to reduce the Hessian oracle for nonconvex-(strongly)-concave problems (Luo et al., 2022; Lin et al., 2020; Yang et al., 2023; Zhang et al., 2022b; Wang et al., 2024b) or study structured nonconvex-nonconcave problems (Zheng et al., 2024; Diakonikolas et al., 2021; Yang et al., 2020; Lee & Kim, 2021; Chen & Luo, 2024).

## ACKNOWLEDGMENTS

Lesi Chen and Jingzhao Zhang are supported by the Shanghai Qi Zhi Institute Innovation Program. Chengchang Liu is supported by the National Natural Science Foundation of China (624B2125).

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
