\|(\boldsymbol{z}_{t+1/2} - \boldsymbol{z}_t) = \boldsymbol{0}, & \text{(Implicit Step)} \\ \boldsymbol{z}_{t+1} = \boldsymbol{z}_t - \dfrac{\boldsymbol{F}(\boldsymbol{z}_{t+1/2})}{M\|\boldsymbol{z}_{t+1/2} - \boldsymbol{z}_t\|}. & \text{(Explicit Step)} \end{cases} \tag{4}$$

The first step (implicit step) solves a cubic regularized sub-problem based on the $\nabla \boldsymbol{F}(\boldsymbol{z}_{\pi(t)})$ computed at the latest snapshot point and $\boldsymbol{F}(\boldsymbol{z}_t)$ at the current iteration point. This step is often viewed as an oracle (Bullins & Lai, 2022; Adil et al., 2022; Lin et al., 2022) as there exists efficient solvers, which will also be discussed in Section 4.3. The second one (explicit step) conducts an extra gradient step based on $\boldsymbol{F}(\boldsymbol{z}_{t+1/2})$.

Reusing the Hessian in the implicit step makes each iteration much cheaper, but would cause additional errors compared to previous methods (Monteiro & Svaiter, 2012; Huang & Zhang, 2022; Adil et al., 2022; Lin et al., 2022). The error resulting from the lazy Hessian updates is formally characterized by the following theorem.

**Lemma 4.1.** *Suppose that Assumption 3.1 and 3.2 hold. For any $\boldsymbol{z} \in \mathbb{R}^d$, Algorithm 1 ensures*

$$\gamma_t^{-1}\langle \boldsymbol{F}(\boldsymbol{z}_{t+1/2}), \boldsymbol{z}_{t+1/2} - \boldsymbol{z} \rangle \leq \frac{1}{2}\|\boldsymbol{z}_t - \boldsymbol{z}\|^2 - \frac{1}{2}\|\boldsymbol{z}_{t+1} - \boldsymbol{z}\|^2 - \frac{1}{2}\|\boldsymbol{z}_{t+1/2} - \boldsymbol{z}_{t+1}\|^2$$
$$- \frac{1}{2}\|\boldsymbol{z}_t - \boldsymbol{z}_{t+1/2}\|^2 + \frac{\rho^2}{2M^2}\|\boldsymbol{z}_t - \boldsymbol{z}_{t+1/2}\|^2 + \underbrace{\frac{2\rho^2}{M^2}\|\boldsymbol{z}_{\pi(t)} - \boldsymbol{z}_t\|^2}_{(*)}.$$

Above, (*) is the error from lazy Hessian updates. Note that (*) vanishes when the current Hessian is used. For lazy Hessian updates, the error would accumulate in the epoch.

The key step in our analysis shows that we can use the negative terms in the right-hand side of the inequality in Lemma 4.1 to bound the accumulated error by choosing $M$ sufficiently large, with the help of the following technical lemma.

**Lemma 4.2.** *For any sequence of positive numbers $\{r_t\}_{t \geq 0}$, it holds for any $m \geq 2$ that* $\sum_{t=1}^{m-1} \left( \sum_{i=0}^{t-1} r_i \right)^2 \leq \frac{m^2}{2} \sum_{t=0}^{m-1} r_t^2$.

When $m = 1$, the algorithm reduces to the NPE algorithm (Monteiro & Svaiter, 2012; Bullins & Lai, 2022; Adil et al., 2022; Lin et al., 2022) without using lazy Hessian updates. When $m \geq 2$, we use Lemma 4.2 to upper bound the error that arises from lazy Hessian updates. Finally, we prove the following guarantee for our proposed algorithm.

**Theorem 4.1** (C-C setting). *Suppose that Assumption 3.1 and 3.2 hold. Let $\boldsymbol{z}^* = (\boldsymbol{x}^*, \boldsymbol{y}^*)$ be a saddle point and $\beta = \|\boldsymbol{z}_0 - \boldsymbol{z}^*\|$. Set $M \geq 3\rho m$. The sequence of iterates generated by Algorithm 1 is bounded $\boldsymbol{z}_t \in \mathbb{B}_\beta(\boldsymbol{z}^*)$, $\boldsymbol{z}_{t+1/2} \in \mathbb{B}_{3\beta}(\boldsymbol{

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

## A  Some Useful Lemmas

**Lemma A.1.** *Recall the definition of restricted gap function in Definition 3.4. For any point $(\hat{\boldsymbol{x}}, \hat{\boldsymbol{y}})$ We have that*

$$\text{Gap}(\hat{\boldsymbol{x}}, \hat{\boldsymbol{y}}; \beta) \leq \max_{\boldsymbol{z} \in \mathbb{B}_{\sqrt{2}\beta(\boldsymbol{z}^*)}} \{f(\hat{\boldsymbol{x}}, \boldsymbol{y}) - f(\boldsymbol{x}, \hat{\boldsymbol{y}})\}, \quad \boldsymbol{z} = (\boldsymbol{x}, \boldsymbol{y}).$$

*Proof.* By Definition 3.4, we have that

$$\text{Gap}(\hat{\boldsymbol{x}}, \hat{\boldsymbol{y}}; \beta) = \max_{\boldsymbol{x} \in \mathbb{B}_\beta(\boldsymbol{x}^*), \boldsymbol{y} \in \mathbb{B}_\beta(\boldsymbol{y}^*)} \{f(\hat{\boldsymbol{x}}, \boldsymbol{y}) - f(\boldsymbol{x}, \hat{\boldsymbol{y}})\} \leq \max_{\boldsymbol{z} \in \mathbb{B}_{\sqrt{2}\beta(\boldsymbol{z}^*)}} \{f(\hat{\boldsymbol{x}}, \boldsymbol{y}) - f(\boldsymbol{x}, \hat{\boldsymbol{y}})\}.$$

$\square$

**Lemma A.2** (Proposition 2.8 Lin et al. (2022))**.** *Let*

$$\bar{\boldsymbol{x}}_t = \frac{1}{\sum_{i=0}^t \eta_i} \sum_{i=0}^t \eta_i \boldsymbol{x}_i, \quad \bar{\boldsymbol{y}}_t = \frac{1}{\sum_{i=0}^t \eta_i} \sum_{i=0}^t \eta_i \boldsymbol{y}_i.$$

*Then under Assumption 3.2, for any $\boldsymbol{z} = (\boldsymbol{x}, \boldsymbol{y})$, it holds that*

$$f(\bar{\boldsymbol{x}}_t, \boldsymbol{y}) - f(\boldsymbol{x}, \bar{\boldsymbol{y}}_t) \leq \frac{1}{\sum_{i=0}^t \eta_i} \sum_{i=0}^t \eta_i \langle \boldsymbol{F}(\boldsymbol{z}_i), \boldsymbol{z}_i - \boldsymbol{z} \rangle.$$

## B  Proof of Lemma 4.2

*Proof.* We prove the result by induction.

Apparently, it is true for $m = 2$, which is the induction base.

Assume that it holds for $m \geq 2$. Then

$$
\begin{aligned}
&\sum_{t=1}^m \left( \sum_{i=0}^{t-1} r_i \right)^2 \\
&= \sum_{t=1}^{m-1} \left( \sum_{i=0}^{t-1} r_i \right)^2 + \left( \sum_{i=0}^{m-1} r_i \right)^2 \\
&\leq \frac{m^2}{2} \sum_{t=0}^{m-1} r_t^2 + m \sum_{t=0}^{m-1} r_t^2 \\
&\leq \left( \frac{m^2 + 2m}{2} \right) \sum_{t=0}^{m-1} r_t^2 \\
&\leq \frac{(m+1)^2}{2} \sum_{t=0}^{m-1} r_t^2.
\end{aligned}
$$

$\square$

## C  Proof of Lemma 4.1

*Proof.* Instead of directly providing a proof for Algorithm 1, we give the proof for the more general inexact algorithm (Algorithm 4), which recovers Algorithm 1 if $\alpha = 1$.

For convenience, we denote $\eta_t = 1/\gamma_t$.

For ant $z \in \mathbb{R}^d$, we have

$$\eta_t \langle \boldsymbol{F}(z_{t+1/2}), z_{t+1/2} - z \rangle$$
$$= \langle z_t - z_{t+1}, z_{t+1/2} - z \rangle$$
$$= \langle z_t - z_{t+1}, z_{t+1} - z \rangle + \langle z_t - z_{t+1}, z_{t+1/2} - z_{t+1} \rangle$$
$$= \langle z_t - z_{t+1}, z_{t+1} - z \rangle + \langle z_t - z_{t+1/2}, z_{t+1/2} - z_{t+1} \rangle + \langle z_{t+1/2} - z_{t+1}, z_{t+1/2} - z_{t+1} \rangle$$
$$= \frac{1}{2}\|z_t - z\|^2 - \frac{1}{2}\|z_{t+1} - z\|^2 - \frac{1}{2}\|\cancel{z_t - z_{t+1}}\|^2$$
$$+ \frac{1}{2}\|\cancel{z_t - z_{t+1}}\|^2 - \frac{1}{2}\|z_{t+1/2} - z_{t+1}\|^2 - \frac{1}{2}\|z_t - z_{t+1/2}\|^2 + \|z_{t+1/2} - z_{t+1}\|^2. \tag{13}$$

Note that by the updates of the algorithm, we have that

$$\gamma_t(z_t - z_{t+1/2}) = \boldsymbol{F}(z_t) + \nabla \boldsymbol{F}(z_{\pi(t)})(z_{t+1/2} - z_t),$$
$$\gamma_t(z_t - z_{t+1}) = \boldsymbol{F}(z_{t+1/2}).$$

It implies that

$$z_{t+1/2} - z_{t+1}$$
$$= \eta_t(\boldsymbol{F}(z_{t+1/2}) - \boldsymbol{F}(z_t) - \nabla \boldsymbol{F}(z_{\pi(t)})(z_{t+1/2} - z_t)))$$
$$= \eta_t(\boldsymbol{F}(z_{t+1/2}) - \boldsymbol{F}(z_t) - \nabla \boldsymbol{F}(z_t)(z_{t+1/2} - z_t)) + \eta_t(\nabla \boldsymbol{F}(z_{\pi(t)})) - \nabla \boldsymbol{F}(z_t))(z_{t+1/2} - z_t) \tag{14}$$

Note that $\nabla \boldsymbol{F}$ is $\rho$-Lipschitz continuous. Taking norm on both sides of (14), we have that

$$\|z_{t+1/2} - z_{t+1}\| \le \frac{\rho \eta_t}{2}\|z_{t+1/2} - z_t\|^2 + \rho \eta_t \|z_{\pi(t)} - z_t\|\|z_{t+1/2} - z_t\|$$
$$\le \frac{\rho}{2M}\|z_{t+1/2} - z_t\| + \frac{\rho}{M}\|z_{\pi(t)} - z_t\|,$$

where we use the condition $M\|z_t - z_{t+1/2}\| \le \gamma_t$ in the last step.

By Young's inequality, this further means

$$\|z_{t+1/2} - z_{t+1}\|^2 \le \frac{\rho^2}{2M^2}\|z_{t+1/2} - z_t\|^2 + \frac{2\rho^2}{M^2}\|z_{\pi(t)} - z_t\|^2.$$

Plug the above inequality into the last term in (13).

$$\eta_t \langle \boldsymbol{F}(z_{t+1/2}), z_{t+1/2} - z \rangle$$
$$\le \frac{1}{2}\|z_t - z\|^2 - \frac{1}{2}\|z_{t+1} - z\|^2 - \frac{1}{2}\|z_{t+1/2} - z_{t+1}\|^2$$
$$- \frac{1}{2}\|z_t - z_{t+1/2}\|^2 + \frac{\rho^2}{2M^2}\|z_t - z_{t+1/2}\|^2 + \frac{2\rho^2}{M^2}\|z_{\pi(t)} - z_t\|^2.$$

$\square$

## D    PROOF OF THEOREM 4.1

*Proof.* When $m = 1$, the algorithm reduces to the NPE algorithm (Monteiro & Svaiter, 2012; Bullins & Lai, 2022; Adil et al., 2022; Lin et al., 2022). When $m \ge 2$, we use Lemma 4.2 to bound the error that arises from lazy Hessian updates.

Instead of directly providing a proof for Algorithm 1, we give the proof for the more general inexact algorithm (Algorithm 4), which recovers Algorithm 1 if $\alpha = 1$.

Define $r_t = \|z_{t+1} - z_t\|$. By triangle inequality and Young's inequality, we have

$$\eta_t \langle \boldsymbol{F}(z_{t+1/2}), z_{t+1/2} - z \rangle$$
$$\le \frac{1}{2}\|z_t - z\|^2 - \frac{1}{2}\|z_{t+1} - z\|^2 - \left(\frac{1}{4} - \frac{\rho^2}{2M^2}\right)\|z_t - z_{t+1/2}\|^2$$
$$- \left(\frac{1}{8}r_t^2 - \frac{2\rho^2}{M^2}\left(\sum_{i=\pi(t)}^{t-1} r_i\right)^2\right).$$

For any $1 \le s \le m$. Telescoping over $t = \pi(k), \cdots, \pi(k) + s - 1$, we have

$$\sum_{t=\pi(k)}^{\pi(k)+s-1} \eta_t \langle \boldsymbol{F}(\boldsymbol{z}_{t+1/2}), \boldsymbol{z}_{t+1/2} - \boldsymbol{z} \rangle$$

$$\le \frac{1}{2} \|\boldsymbol{z}_{\pi(k)} - \boldsymbol{z}\|^2 - \frac{1}{2} \|\boldsymbol{z}_{\pi(k)+s} - \boldsymbol{z}\|^2 - \left( \frac{1}{4} - \frac{\rho^2}{2M^2} \right) \sum_{t=\pi(k)}^{\pi(k)+s-1} \|\boldsymbol{z}_t - \boldsymbol{z}_{t+1/2}\|^2$$

$$- \left( \frac{1}{8} \sum_{t=\pi(k)}^{\pi(k)+s-1} r_t^2 - \frac{2\rho^2}{M^2} \sum_{t=\pi(k)+1}^{\pi(k)+s-1} \left( \sum_{i=\pi(k)}^{t-1} r_i \right)^2 \right).$$

Applying Lemma 4.2, we further have

$$\sum_{t=\pi(k)}^{\pi(k)+s-1} \eta_t \langle \boldsymbol{F}(\boldsymbol{z}_{t+1/2}), \boldsymbol{z}_{t+1/2} - \boldsymbol{z} \rangle$$

$$\le \frac{1}{2} \|\boldsymbol{z}_{\pi(k)} - \boldsymbol{z}\|^2 - \frac{1}{2} \|\boldsymbol{z}_{\pi(k)+s} - \boldsymbol{z}\|^2 - \left( \frac{1}{4} - \frac{\rho^2}{2M^2} \right) \sum_{t=\pi(k)}^{\pi(k)+s-1} \|\boldsymbol{z}_t - \boldsymbol{z}_{t+1/2}\|^2$$

$$- \left( \frac{1}{8} - \frac{\rho^2 s^2}{M^2} \right) \sum_{t=\pi(k)}^{\pi(k)+s-1} r_t^2.$$

Note that $s \le m$. Let $M \ge 3\rho m$. Then

$$\sum_{t=\pi(k)}^{\pi(k)+s-1} \eta_t \langle \boldsymbol{F}(\boldsymbol{z}_{t+1/2}), \boldsymbol{z}_{t+1/2} - \boldsymbol{z} \rangle$$

$$\le \frac{1}{2} \|\boldsymbol{z}_{\pi(k)} - \boldsymbol{z}\|^2 - \frac{1}{2} \|\boldsymbol{z}_{\pi(k)+s} - \boldsymbol{z}\|^2 - \frac{1}{8} \sum_{t=\pi(k)}^{\pi(k)+s-1} \|\boldsymbol{z}_t - \boldsymbol{z}_{t+1/2}\|^2.$$

Let $s = m$ and further telescope over $k = 0, m, 2m, \cdots$. Then

$$\sum_{t=0}^{T} \eta_t \langle \boldsymbol{F}(\boldsymbol{z}_{t+1/2}), \boldsymbol{z}_{t+1/2} - \boldsymbol{z} \rangle \le \frac{1}{2} \|\boldsymbol{z}_0 - \boldsymbol{z}\|^2 - \frac{1}{2} \|\boldsymbol{z}_T - \boldsymbol{z}\|^2 - \frac{1}{8} \sum_{t=0}^{T} \|\boldsymbol{z}_t - \boldsymbol{z}_{t+1/2}\|^2. \quad (15)$$

This inequality is the key to the convergence. It implies the following results. First, letting $\boldsymbol{z} = \boldsymbol{z}^*$ and using the fact that $\langle \boldsymbol{F}(\boldsymbol{z}_{t+1/2}), \boldsymbol{z}_{t+1/2} - \boldsymbol{z}^* \rangle \ge 0$ according to monotonicity of $\boldsymbol{F}$, we can prove the iterate is bounded

$$\|\boldsymbol{z}_t - \boldsymbol{z}^*\| \le \|\boldsymbol{z}_0 - \boldsymbol{z}^*\|, \quad \text{and} \quad \|\boldsymbol{z}_t - \boldsymbol{z}_{t+1/2}\| \le 2\|\boldsymbol{z}_0 - \boldsymbol{z}^*\|, \quad t = 0, \cdots, T - 1. \quad (16)$$

Then using triangle inequality, we obtain

$$\|\boldsymbol{z}_{t+1/2} - \boldsymbol{z}^*\| \le 3\|\boldsymbol{z}_0 - \boldsymbol{z}^*\|, \quad \forall t = 0, \cdots, T - 1.$$

Second, beginning with Lemma A.1 we have that

$$\text{Gap}(\bar{\boldsymbol{x}}_T, \bar{\boldsymbol{y}}_T; 3\beta) \le \max_{\boldsymbol{z} \in \mathbb{B}_{3\sqrt{2}\beta}(\boldsymbol{z}^*)} \{ f(\bar{\boldsymbol{x}}_T, \boldsymbol{y}) - f(\boldsymbol{x}, \bar{\boldsymbol{y}}_T) \}$$

$$\overset{(a)}{\le} \max_{\boldsymbol{z} \in \mathbb{B}_{3\sqrt{2}\beta}(\boldsymbol{z}^*)} \left\{ \frac{1}{\sum_{t=0}^{T} \eta_t} \sum_{t=0}^{T} \eta_t \langle \boldsymbol{F}(\boldsymbol{z}_{t+1/2}), \boldsymbol{z}_{t+1/2} - \boldsymbol{z} \rangle \right\} \quad (17)$$

$$\overset{(b)}{\le} \frac{\max_{\boldsymbol{z} \in \mathbb{B}_{3\sqrt{2}\beta}(\boldsymbol{z}^*)} \{ \|\boldsymbol{z}_0 - \boldsymbol{z}\|^2 \}}{2\sum_{t=0}^{T-1} \eta_t} \overset{(c)}{\le} \frac{16\|\boldsymbol{z}_0 - \boldsymbol{z}^*\|^2}{\sum_{t=0}^{T-1} \eta_t},$$

where (a) uses Lemma A.2, (b) uses (15) and (c) uses the fact that $z \in \mathbb{B}_{3\sqrt{2}\beta}(z^*)$. Third, we can also use (15) to lower bound $\sum_{t=0}^{T-1} \eta_t$. (15) with $z = z^*$ implies

$$\sum_{t=0}^{T} \gamma_t^2 \leq 4\alpha^2 M^2 \|z_0 - z^*\|^2,$$

where we use the condition $\gamma_t \leq \alpha M \|z_t - z_{t+1/2}\|$ in the last step. Then by Holder's inequality,

$$T = \sum_{t=0}^{T-1} (\eta_t)^{2/3} (\gamma_t^2)^{1/3} \leq \left( \sum_{t=0}^{T-1} \eta_t \right)^{2/3} \left( \sum_{t=0}^{T-1} \gamma_t^2 \right)^{1/3}.$$

Therefore,

$$\sum_{t=0}^{T-1} \eta_t \geq \frac{T^{3/2}}{2\alpha M \|z_0 - z^*\|}. \tag{18}$$

We plug in (18) to (17) and obtain that

$$\text{Gap}(\bar{x}_T, \bar{y}_T; 3\beta) \leq \frac{32\alpha M \|z_0 - z^*\|^3}{T^{3/2}}.$$

The desired theorem is the case $\alpha = 1$. $\qquad \square$

## E  PROOF OF THEOREM 4.2

*Proof.* Using the strongly monotonicity of operator $F$ in (15), we obtain that

$$\sum_{t=0}^{T} \mu \eta_t \|z_{t+1/2} - z^*\|^2 \leq \frac{1}{2} \|z_0 - z^*\|^2 - \frac{1}{2} \|z_T - z^*\|^2.$$

Using Jensen's inequality, for each epoch, we have

$$\|\bar{z}_T - z^*\|^2 \leq \frac{\|z_0 - z^*\|^2}{2\mu \sum_{t=0}^{T-1} \eta_t} \leq \frac{M \|z_0 - z^*\|^3}{\mu T^{3/2}} := c \|z_0 - z^*\|^2.$$

Next, we consider the iterate $\{z^{(s)}\}_{s=0}^{S-1}$. For the first epoch, the setting of $T$ ensures $c \leq 1/2$:

$$\|z^{(1)} - z^*\|^2 \leq \frac{1}{2} \|z_0 - z^*\|^2.$$

Then for the second one, it is improved by

$$\|z^{(2)} - z^*\|^2 \leq \frac{\|z^{(1)} - z^*\|^3}{2\|z_0 - z^*\|} \leq \left(\frac{1}{2}\right)^{1+3/2} \|z_0 - z^*\|^2.$$

Keep repeating this process. We can get

$$\|z^{(s)} - z^*\|^2 \leq \left(\frac{1}{2}\right)^{q_s} \|z_0 - z^*\|^2,$$

where $q_s$ satisfies the recursion

$$q_s = \begin{cases} 1, & s = 1; \\ \frac{3}{2} q_{s-1} + 1, & s \geq 2. \end{cases}$$

This implies

$$\|z^{(s)} - z^*\|^2 \leq \left(\frac{1}{2}\right)^{\left(\frac{3}{2}\right)^{s-1}+1} \|z_0 - z^*\|^2.$$

---

**Algorithm 4** Inexact LEN$(z_0, T, m, M, \alpha)$

---

1: **for** $t = 0, \cdots, T-1$ **do**
2:     Use Algorithm 5 to find $(z_{t+1/2}, \gamma_t)$ that satisfies

$$z_{t+1/2} = z_t - (\nabla F(z_{\pi(t)}) + \gamma_t I_d)^{-1} F(z_t)$$

    and $M\|z_t - z_{t+1/2}\| \leq \gamma_t \leq \alpha M\|z_t - z_{t+1/2}\|$ for given $\alpha \geq 1$.
3:     Compute extra-gradient step $z_{t+1} = z_t - \gamma_t^{-1} F(z_{t+1/2})$.
4: **end for**
5: **return** $\bar{z}_T = \frac{1}{\sum_{t=0}^{T-1} \gamma_t^{-1}} \sum_{t=0}^{T-1} \gamma_t^{-1} z_{t+1/2}$.

---

Set $m = \Theta(d)$, LEN-restart takes $\mathcal{O}(d^{2/3}\kappa^{2/3} \log\log(1/\epsilon))$ oracle to $F(\cdot)$ and $\mathcal{O}((1 + d^{-1/3}\kappa^{2/3}) \log\log(1/\epsilon))$ oracle to $\nabla F(\cdot)$. Under Assumption 3.4, the computational complexities of the oracles is

$$\mathcal{O}\left(N \cdot d^{2/3}\kappa^{2/3} \log\log(1/\epsilon) + Nd \cdot (1 + d^{-1/3}\kappa^{2/3}) \log\log(1/\epsilon)\right)$$
$$= \mathcal{O}\left((Nd + Nd^{2/3}\kappa^{2/3}) \log\log(1/\epsilon)\right).$$

$\square$

## F    PROOF OF COROLLARY 4.1

*Proof.* The computational complexity of inner loop can be directly obtained by replacing $\epsilon^{-1}$ by $\kappa$ in Theorem 4.3 such that

$$\text{Inner Computational Complexity} = \tilde{\mathcal{O}}\left((N + d^2) \cdot (d + d^{2/3}\kappa^{2/3})\right).$$

The iterations of outer loop is $S = \log\log(1/\epsilon)$, thus, the total computational complexity of LEN-restart is

$$S \cdot \text{Inner Computational Complexity} = \tilde{\mathcal{O}}\left((N + d^2) \cdot (d + d^{2/3}\kappa^{2/3})\right).$$

$\square$

## G    COMPUTATIONAL COMPLEXITY USING FAST MATRIX OPERATIONS

Theoretically, one may use fast matrix operations for Schur decomposition and matrix inversion (Demmel et al., 2007), with a computational complexity of $d^\omega$, where $\omega \approx 2.371552$ is the matrix multiplication constant. In this case, the total computational complexity of Algorithm 3 is

$$\tilde{\mathcal{O}}\left(\left(\frac{Nd + d^\omega}{m} + d^2 + N\right) m^{2/3}\epsilon^{-2/3}\right)$$

Setting the optimal $m$, we obtain the following complexity of Algorithm 3:

$$\begin{cases} \tilde{\mathcal{O}}(d^{\frac{2}{3}(\omega+1)}\epsilon^{-2/3}) & (\text{with } m = d^{\omega-2}), & N \lesssim d^{\omega-1} \\ \tilde{\mathcal{O}}(N^{2/3}d^{4/3}\epsilon^{-2/3}) & (\text{with } m = N/d), & d^{\omega-1} \lesssim N \lesssim d^2 \\ \tilde{\mathcal{O}}(Nd^{2/3}\epsilon^{-2/3}) & (\text{with } m = d), & d^2 \lesssim N. \end{cases}$$

Our result is always better than the $\mathcal{O}((Nd + d^\omega)\epsilon^{-2/3})$ of existing optimal second-order methods.

## H    THE INEXACT ALGORITHM

Algorithm 1 requires a cubic regularized Newton (CRN) oracle (Implicit Step, (4)). We provide implementation details for the CRN oracle in Section 4.3. One missing detail is that we can not

obtain the exact solution to the CRN oracle in practice. To make our result more rigorous, we analyze the inexact LEN (Algorithm 1), which allows inexact sub-problem solving with a parameter $\alpha \geq 1$. Note that this algorithm reduces to the exact version (Algorithm 1) when $\alpha = 1$.

Below, we present the following theorem as the inexact version of Theorem 4.1.

**Theorem H.1.** *Suppose that Assumption 3.1 and 3.2 hold. Let $\boldsymbol{z}^* = (\boldsymbol{x}^*, \boldsymbol{y}^*)$ be a saddle point and $\beta = \|\boldsymbol{z}_0 - \boldsymbol{z}^*\|$. Set $M \geq 3\rho m$. The sequence of iterates generated by Algorithm 4 is bounded $\boldsymbol{z}_t \in \mathbb{B}_\beta(\boldsymbol{z}^*)$, $\boldsymbol{z}_{t+1/2} \in \mathbb{B}_{3\beta}(\boldsymbol{z}^*)$, $\quad \forall t = 0, \cdots, T-1$, and satisfies the following ergodic convergence:*

$$\mathrm{Gap}(\bar{\boldsymbol{x}}_T, \bar{\boldsymbol{y}}_T; 3\beta) \leq \frac{16\alpha M\|\boldsymbol{z}_0 - \boldsymbol{z}^*\|^3}{T^{3/2}}.$$

*Let $M = 3\rho m$ and $\alpha = 2$. Algorithm 1 finds an $\epsilon$-saddle point within $\mathcal{O}(m^{2/3}\epsilon^{-2/3})$ iterations.*

*Proof.* See Section D. □

The only remaining thing is to show how to compute $\gamma_t$ in the auxiliary problem (Line 2 in Algorithm 4). Below, we present an efficient sub-procedure to achieve the desired goal using the standard Newton step. We define the monotone operator $\boldsymbol{A}_t : \mathbb{R}^d \to \mathbb{R}^d$ by

$$\boldsymbol{A}_t(\boldsymbol{z}) = \boldsymbol{F}(\boldsymbol{z}_t) + \nabla \boldsymbol{F}(\boldsymbol{z}_{\pi(t)})(\boldsymbol{z} - \boldsymbol{z}_t). \tag{19}$$

Then we can write down the (regularized) Newton step as

$$\begin{aligned} \boldsymbol{z}_{t+1/2}(\eta; \boldsymbol{z}_t) &:= \boldsymbol{z}_t - (\nabla \boldsymbol{F}(\boldsymbol{z}_{\pi(t)}) + \eta^{-1}\boldsymbol{I}_d)^{-1}\boldsymbol{F}(\boldsymbol{z}_t) \\ &= (\boldsymbol{I}_d + \eta\boldsymbol{A}_t)^{-1}(\boldsymbol{z}_t). \end{aligned} \tag{20}$$

And the inexact condition (Line 2 in Algorithm 4) is

$$\frac{1}{\alpha M} \leq \phi_t(\eta; \boldsymbol{z}_t) \leq \frac{1}{M}, \tag{21}$$

where $\phi_t(\eta; \boldsymbol{z}_t)$ is defined as $\phi_t(\eta; \boldsymbol{z}_t) := \eta\|\boldsymbol{z}_{t+1/2}(\eta; \boldsymbol{z}_t) - \boldsymbol{z}_t\|$.

Note that a stepsize $\eta$ that satisfies (21) directly implies $\gamma_t = 1/\eta$ satisfies the requirement of Line 2 in Algorithm 4. Therefore, the main goal of this section is to design a sub-procedure that can determine the stepsize $\eta$ that satisfies (21).

A similar sub-procedure without using lazy Hessian updates has been proposed in (Monteiro & Svaiter, 2012). Below, we show that we can use a similar sub-procedure for our algorithm. We recall some useful lemmas in (Monteiro & Svaiter, 2012), which holds for any monotone operators $\boldsymbol{A}$. Below, we state their results when $\boldsymbol{A} = \boldsymbol{A}_t$.

**Lemma H.1** (Lemma 4.3 and Lemma 4.4 (Monteiro & Svaiter, 2012)). *Recall the definition of $\phi_t$ right after (21). For any $\boldsymbol{z} \in \mathbb{R}^d$, the following statements hold:*

1. *For any $\eta > 0$, we have $\phi_t(\eta; \boldsymbol{z}) > 0$.*

2. *For any $0 < \eta' \leq \eta$, we have that*

$$\frac{\eta}{\eta'}\phi_t(\eta'; \boldsymbol{z}) \leq \phi_t(\eta; \boldsymbol{z}) \leq \left(\frac{\eta}{\eta'}\right)^2 \phi_t(\eta'; \boldsymbol{z}).$$

   *As a corollary, $\phi_t(\eta; \boldsymbol{z})$ is a continuous and strictly increasing function, which converges to 0 or $+\infty$ as $\eta$ tends to 0 or $+\infty$, respectively.*

3. *For any $0 < \beta^- < \beta^+$, the set of all scalars $\eta > 0$ satisfying $\beta^- \leq \phi_t(\eta; \boldsymbol{z}) \leq \beta^+$ is a closed interval $[\eta^-, \eta^+]$ such that $\eta^+/\eta^- \geq \sqrt{\beta^+/\beta^-}$.*

Algorithm 5 presents our sub-procedure to output the tuple $(\boldsymbol{z}_{t+1/2}, \gamma_t)$ satisfying (21). Similar to (Monteiro & Svaiter, 2012), the procedure consists of two stages. The first one is a bracketing stage, which either outputs an acceptable solution or an initial interval $[c_t^-, c_t^+]$ that contains all the $\eta$ satisfying (21). The second one is a bisection stage, which uses binary search in the logarithmic scale to find a stepsize $\eta$ satisfying (21). Note that the log-scale binary search would finally lead to a $\mathcal{O}(\log\log(1/\epsilon))$ iteration complexity, which improves the $\mathcal{O}(\log(1/\epsilon))$ iteration complexity using naive binary search in (Adil et al., 2022; Bullins & Lai, 2022).

Our first result of Algorithm 5 is the correctness of the bracketing stage, stated as follows.

---

**Algorithm 5** Bracketing/Bisection Procedure$(\boldsymbol{A}_t, \boldsymbol{z}_t, M, \alpha, \eta_t^0)$

---

1: **(Bracketing Stage)** Compute $\boldsymbol{z}_{t+1/2}^0 = (\boldsymbol{I}_d + \eta_t^0 \boldsymbol{A}_t)^{-1}(\boldsymbol{z}_t)$ with one Newton step.

    (1a) **if** $\eta_t^0 \|\boldsymbol{z}_{t+1/2}^0 - \boldsymbol{z}_t\| \in (\frac{1}{\alpha M}, \frac{1}{M})$, **then** let $\boldsymbol{z}_{t+1/2} = \boldsymbol{z}_{t+1/2}^0, \eta_t = \eta_t^0$ and **go to** Line 3.

    (1b) **if** $\eta_t^0 \|\boldsymbol{z}_{t+1/2}^0 - \boldsymbol{z}_t\| < \frac{1}{\alpha M}$, **then** set $c_t^- = \eta_t^0$ and $c_t^+ = \frac{1}{M\|\boldsymbol{z}_{t+1/2}^0 - \boldsymbol{z}_t\|}$;

    (1c) **if** $\eta_0^t \|\boldsymbol{z}_{t+1/2}^0 - \boldsymbol{z}_t\| > \frac{1}{M}$, **then** set $c_t^- = \frac{1}{\alpha M\|\boldsymbol{z}_{t+1/2}^0 - \boldsymbol{z}_t\|}$ and $c_t^+ = \eta_t^0$;

2: **(Bisection Stage)**

    (2a) set $\eta_t = \sqrt{c_t^- c_t^+}$ and compute $\boldsymbol{z}_{t+1/2} = (\boldsymbol{I}_d + \eta_t \boldsymbol{A}_t)^{-1}(\boldsymbol{z}_t)$ with one Newton step;

    (2b) **if** $\eta_t \|\boldsymbol{z}_{t+1/2} - \boldsymbol{z}_t\| \in (\frac{1}{\alpha M}, \frac{1}{M})$, **then go to** Line 3;

    (2c) **if** $\eta_t \|\boldsymbol{z}_{t+1/2} - \boldsymbol{z}_t\| > \frac{1}{M}$, **then** set $c_t^+ = \eta_t$; **else** set $c_t^- = \eta_t$;

    (2d) **go to** step (2a).

3: **return** $(\boldsymbol{z}_{t+1/2}, \gamma_t)$ that meets the requirement of Line 2 in Algorithm 4, where $\gamma_t = 1/\eta_t$ .

---

**Lemma H.2.** *Let $[\eta_t^-, \eta_t^+]$ be the interval that contains all the stepsizes satisfying (21). Compute $\boldsymbol{z}_{t+1/2}^0 = (\boldsymbol{I}_d + \eta_t^0 \boldsymbol{A}_t)^{-1}(\boldsymbol{z}_t)$ with one Newton step as Algorithm 5. The following statements hold:*

*1. if $\eta_t^0 \|\boldsymbol{z}_{t+1/2}^0 - \boldsymbol{z}_t\| < \frac{1}{\alpha M}$, then $\eta_t^0 < \eta_t^-$ and $\eta_t^+ \leq \frac{1}{M\|\boldsymbol{z}_{t+1/2}^0 - \boldsymbol{z}_t\|}$;*

*2. if $\eta_t^0 \|\boldsymbol{z}_{t+1/2}^0 - \boldsymbol{z}_t\| > \frac{1}{M}$, then $\eta_t^+ < \eta_t^0$ and $\frac{1}{\alpha M\|\boldsymbol{z}_{t+1/2}^0 - \boldsymbol{z}_t\|} \leq \eta_t^-$.*

*Proof.* We only prove the first claim since the proof of the second claim follows in a similar manner.

Recall the definition of $\phi_t$ right after (21). The condition $\eta_t^0 \|\boldsymbol{z}_{t+1/2}^0 - \boldsymbol{z}_t\| < \frac{1}{\alpha M}$ is equivalent to $\phi_t(\eta_t^0; \boldsymbol{z}_t) < \phi_t(\eta_t^-; \boldsymbol{z}_t)$. Firstly. the fact that $\phi_t(\eta_t^-; \boldsymbol{z}_t)$ is a strictly increasing function according to the second statement in Lemma H.1, we know that $\eta_t^0 < \eta_t^-$ .

Secondly, using the inequality in the second statement of Lemma H.1, we know that

$$\eta_t^+ \|\boldsymbol{z}_{t+1/2}^0 - \boldsymbol{z}_t\| = \frac{\eta_t^+}{\eta_t^0} \phi_t(\eta_t^0; \boldsymbol{z}_t) \leq \phi_t(\eta_t^+; \boldsymbol{z}_t) = \frac{1}{M},$$

which implies $\eta_t^+ \leq \frac{1}{M\|\boldsymbol{z}_{t+1/2}^0 - \boldsymbol{z}_t\|}$ by rearranging.

$\square$

Therefore, the bracketing stage can always output an interval that contains the acceptable stepsizes $\eta$ satisfying (21). Given such a valid initial interval, the bisection stage always find an acceptable stepsize, stated as follows.

**Lemma H.3.** *Consider Algorithm 5. If the bracketing stage outputs an interval $[c_t^-, c_t^+]$ containing all the stepsizes $\eta$ satisfying (21), which is then input to the bisection stage, then the number of Newton step during the bisection stage is bounded by $1 + \log(\log(h_t)/\log \alpha)$, where*

$$h_t = \max \left\{ \frac{1}{\eta_t^0 M \|\boldsymbol{z}_{t+1/2}^0 - \boldsymbol{z}_t\|}, \; \alpha M \eta_t^0 \|\boldsymbol{z}_{t+1/2}^0 - \boldsymbol{z}_t\| \right\} \tag{22}$$

*is the maximal ratio of $c_t^+ / c_t^-$.*

*Proof.* After $j$ steps of bisection iterations, we have that $\log \frac{c_t^+}{c_t^-} = \frac{1}{2^j} \log h_t$. In view of the third statement in Lemma H.1, we know that $c_t^+ / c_t^- \geq \sqrt{\alpha}$. These two inequalities immediately imply that the bisection stage would terminates in $j \leq 1 + \log(\log(h_t)/\log \alpha)$ iterations. $\square$

Our goal from now on would be giving a uniform upper bound of $h_t$ all for $t$, which can imply the total complexity of our algorithm. From the definition of $h_t$ in (22), we need to give both lower and upper bounds of $\eta_t^0 \|\boldsymbol{z}_{t+1/2}^0 - \boldsymbol{z}_t\|$. We recall some technical lemmas in (Monteiro & Svaiter, 2012).

**Lemma H.4** (Proposition 4.5 Monteiro & Svaiter (2012)). *Let $\boldsymbol{A} : \mathbb{R}^d \to \mathbb{R}^d$ be a monotone operator. For a point $\boldsymbol{z}^* \in \mathbb{R}^d$ such that $\boldsymbol{A}(\boldsymbol{z}^*) = 0$, for any $\eta > 0$ and $\boldsymbol{z} \in \mathbb{R}^d$ it holds that*

$$\max \left\{ \|(\boldsymbol{I}_d + \eta \boldsymbol{A})^{-1} \boldsymbol{z} - \boldsymbol{z}^*\|, \|(\boldsymbol{I}_d + \eta \boldsymbol{A})^{-1} \boldsymbol{z} - \boldsymbol{z}\| \right\} \leq \|\boldsymbol{z} - \boldsymbol{z}^*\|.$$

From now on, we will fix all the $\eta_t^0$ in all the iterations such that $\eta_t^0 = \bar{\eta}$ and analyze Algorithm 4. The following lemma shows a uniform upper bound of $\|\boldsymbol{z}_{t+1/2}^0 - \boldsymbol{z}_t\|$.

**Lemma H.5** (Upper bound of $\|\boldsymbol{z}_{t+1/2}^0 - \boldsymbol{z}_t\|$). *Suppose that Assumption 3.1 and 3.2 hold. Let $\boldsymbol{z}^* = (\boldsymbol{x}^*, \boldsymbol{y}^*)$ be a saddle point. Set $M = 3\rho m$ as in Theorem 4.1. For all the iterations of Algorithm 4, it holds that*

$$\|\boldsymbol{z}_{t+1/2}^0 - \boldsymbol{z}_t\| \leq \|\boldsymbol{z}_0 - \boldsymbol{z}^*\| + \frac{5\bar{\eta}\rho}{2} \|\boldsymbol{z}_0 - \boldsymbol{z}^*\|^2. \tag{23}$$

*Proof.* Let $r_t := \boldsymbol{F}(\boldsymbol{z}^*) - \boldsymbol{A}_t(\boldsymbol{z}^*)$ and define the operator $\tilde{\boldsymbol{A}}_t$ as $\tilde{\boldsymbol{A}}_t(\boldsymbol{z}) = \boldsymbol{A}_t(\boldsymbol{z}) + r_t$. From the definition of $\tilde{\boldsymbol{A}}_t$ we know that all any $\eta > 0$ and $\boldsymbol{z} \in \mathbb{R}^d$ we have that

$$(\boldsymbol{I}_d + \eta \tilde{\boldsymbol{A}}_t)^{-1}(\boldsymbol{z} + \eta r_t) = (\boldsymbol{I}_d + \eta \boldsymbol{A}_t)^{-1}(\boldsymbol{z}) \tag{24}$$

Now we upper bound $\|\boldsymbol{z}_{t+1/2}^0 - \boldsymbol{z}_t\|$ as follows.

$$
\begin{aligned}
&\|\boldsymbol{z}_{t+1/2}^0 - \boldsymbol{z}_t\| \\
&= \|(\boldsymbol{I}_d + \bar{\eta} \boldsymbol{A}_t)^{-1}(\boldsymbol{z}_t) - \boldsymbol{z}_t\| \\
&= \|(\boldsymbol{I}_d + \bar{\eta} \tilde{\boldsymbol{A}}_t)^{-1}(\boldsymbol{z}_t + \bar{\eta} r_t) - \boldsymbol{z}_t\| \\
&\leq \|(\boldsymbol{I}_d + \bar{\eta} \tilde{\boldsymbol{A}}_t)^{-1}(\boldsymbol{z}_t) - \boldsymbol{z}_t\| + \|(\boldsymbol{I}_d + \bar{\eta} \tilde{\boldsymbol{A}}_t)^{-1}(\boldsymbol{z}_t) - (\boldsymbol{I}_d + \bar{\eta} \tilde{\boldsymbol{A}}_t)^{-1}(\boldsymbol{z}_t + \bar{\eta} r_t)\| \\
&\leq \|\boldsymbol{z}_t - \boldsymbol{z}^*\| + \bar{\eta}\|r_t\|,
\end{aligned} \tag{25}
$$

where in the last step we use Lemma H.4 to upper bound the first term and use the non-expansiveness of resolvent (see *i.e.* (Rockafellar, 1976)) to upper bound the second term.

We continue to upper bound $\|r_t\|$. Recall the definition of $\boldsymbol{A}_t$ in (19), we know that

$$
\begin{aligned}
r_t &= \boldsymbol{F}(\boldsymbol{z}^*) - \boldsymbol{F}(\boldsymbol{z}_t) - \nabla \boldsymbol{F}(\boldsymbol{z}_{\pi(t)})(\boldsymbol{z}^* - \boldsymbol{z}_t) \\
&= \boldsymbol{F}(\boldsymbol{z}^*) - \boldsymbol{F}(\boldsymbol{z}_t) - \nabla \boldsymbol{F}(\boldsymbol{z}_t)(\boldsymbol{z}^* - \boldsymbol{z}_t) + (\nabla \boldsymbol{F}(\boldsymbol{z}_t) - \nabla \boldsymbol{F}(\boldsymbol{z}_{\pi(t)})(\boldsymbol{z}^* - \boldsymbol{z}_t)
\end{aligned}
$$

Note that $\nabla \boldsymbol{F}$ is $\rho$-Lipschitz continuous. Taking norm on both sides of the above identity, we have

$$\|r_t\| \leq \frac{\rho}{2}\|\boldsymbol{z}^* - \boldsymbol{z}_t\|^2 + \rho\|\boldsymbol{z}_t - \boldsymbol{z}_{\pi(t)}\|\|\boldsymbol{z}^* - \boldsymbol{z}_t\|$$

Recalling (16) that we have $\|\boldsymbol{z}_t - \boldsymbol{z}^*\| \leq \|\boldsymbol{z}_0 - \boldsymbol{z}^*\|$ for all $t$, by the triangle inequality we also have $\|\boldsymbol{z}_t - \boldsymbol{z}_{\pi(t)}\| \leq 2\|\boldsymbol{z}_0 - \boldsymbol{z}^*\|$. Therefore, we have that $\|r_t\| \leq \frac{5}{2}\|\boldsymbol{z}_0 - \boldsymbol{z}^*\|^2$. Finally, we plug into (25) to obtain the desired upper bound in (23).

□

Next, we give a uniform lower bound of $\|\boldsymbol{z}_{t+1/2}^0 - \boldsymbol{z}_t\|$.

**Lemma H.6** (Lower bound of $\|\boldsymbol{z}_{t+1/2}^0 - \boldsymbol{z}_t\|$). *Suppose that Assumption 3.1 and 3.2 hold. Let $\boldsymbol{z}^* = (\boldsymbol{x}^*, \boldsymbol{y}^*)$ be a saddle point and $\beta = \|\boldsymbol{z}_0 - \boldsymbol{z}^*\|$. Set $M = 3\rho m$ as in Theorem 4.1. If in all the iterations of Algorithm 5 the point $\boldsymbol{z}_{t+1/2}^0$ is not an $\epsilon$-solution, it holds that*

$$\bar{\eta}\|\boldsymbol{z}_{t+1/2}^0 - \boldsymbol{z}_t\| \geq \xi_t, \tag{26}$$

*where $\xi_t = \min \left\{ 2\beta, \frac{\bar{\eta}\epsilon}{8\beta(3\bar{\eta}\beta\rho+1)} \right\}$.*

*Proof.* We show a contradiction if (26) does not hold. Firstly, if $\boldsymbol{z}_{t+1/2}^0 = (\boldsymbol{x}_{t+1/2}^0, \boldsymbol{y}_{t+1/2}^0)$ is not an $\epsilon$-solution to the problem, then by Lemma A.1 and A.2 we know that $\|\boldsymbol{F}(\boldsymbol{z}_{t+1/2}^0)\|$ must be large:

$$\epsilon \leq \text{Gap}(\boldsymbol{x}_{t+1/2}^0, \boldsymbol{y}_{t+1/2}^0; 3\beta) \leq \max_{\boldsymbol{z} \in \mathbb{B}_{3\sqrt{2}\beta}(\boldsymbol{z}^*)} \langle \boldsymbol{F}(\boldsymbol{z}_{t+1/2}^0), \boldsymbol{z}_{t+1/2}^0 - \boldsymbol{z} \rangle \leq 8\beta\|\boldsymbol{F}(\boldsymbol{z}_{t+1/2}^0)\|,$$

where the last step uses that $\|z^0_{t+1/2} - z_t\| \leq 2\beta$ if (26) does not hold, $\|z_t - z^*\| \leq \beta$ and the triangle inequality. Therefore, we can conclude that

$$\|\boldsymbol{F}(z^0_{t+1/2})\| \geq \frac{\epsilon}{8\beta}. \tag{27}$$

Secondly, from the update of the algorithm, we have that

$$z_t - z^0_{t+1/2} = \bar{\eta}(\boldsymbol{F}(z_t) + \nabla \boldsymbol{F}(z_{\pi(t)})(z^0_{t+1/2} - z_t))$$

Then we further know that

$$
\begin{aligned}
&z_t - z^0_{t+1/2} - \boldsymbol{F}(z^0_{t+1/2}) \\
&= \bar{\eta}(\boldsymbol{F}(z_t) + \nabla \boldsymbol{F}(z_{\pi(t)})(z^0_{t+1/2} - z_t) - \boldsymbol{F}(z^0_{t+1/2})) \\
&= \bar{\eta}(\boldsymbol{F}(z_t) + \nabla \boldsymbol{F}(z_t)(z^0_{t+1/2} - z_t) - \boldsymbol{F}(z^0_{t+1/2})) \\
&\quad + \bar{\eta}(\nabla \boldsymbol{F}(z_t) - \nabla \boldsymbol{F}(z_{\pi(t)}))(z^0_{t+1/2} - z_t).
\end{aligned}
$$

Note that $\nabla \boldsymbol{F}$ is $\rho$-Lipschitz continuous. Taking norm on both sides of the above identity, we have

$$
\begin{aligned}
&\|z_t - z^0_{t+1/2} - \boldsymbol{F}(z^0_{t+1/2})\| \\
&\leq \frac{\bar{\eta}\rho}{2}\|z^0_{t+1/2} - z_t\|^2 + \bar{\eta}\rho\|z_t - z_{\pi(t)}\|\|z^0_{t+1/2} - z_t\| \\
&\leq 3\bar{\eta}\beta\rho\|z^0_{t+1/2} - z_t\|.
\end{aligned}
$$

where the last step uses the triangle inequality, that $\|z^0_{t+1/2} - z_t\| \leq 2\beta$ if (26) does not hold, and that $\|z_t - z^*\| \leq \|z_0 - z^*\|$ by (16). Then we can know that

$$
\begin{aligned}
\bar{\eta}\|\boldsymbol{F}(z^0_{t+1/2})\| &\leq \|z_t - z^0_{t+1/2} - \bar{\eta}\boldsymbol{F}(z^0_{t+1/2})\| + \|z^0_{t+1/2} - z_t\| \\
&\leq (3\bar{\eta}\beta\rho + 1)\|z^0_{t+1/2} - z_t\|.
\end{aligned}
$$

Recalling (27), we know that this would contradict the hypothesis that (26) does not hold. □

Lemma H.5 and Lemma H.6 tell us that the $h_t$ defined in (22) is uniformly bounded for all $t$. Finally, we obtain the following theorem by combining Theorem H.1 and Theorem H.3.

**Theorem H.2.** *Suppose that Assumption 3.1 and 3.2 hold. Let $z^* = (x^*, y^*)$ be a saddle point and $\beta = \|z_0 - z^*\|$. Set $M \geq 3\rho m$. The sequence of iterates generated by Algorithm 4 is bounded $z_t \in \mathbb{B}_\beta(z^*)$, $z_{t+1/2} \in \mathbb{B}_{3\beta}(z^*)$, $\forall t = 0, \cdots, T-1$, and satisfies the following ergodic convergence:*

$$\mathrm{Gap}(\bar{x}_T, \bar{y}_T; 3\beta) \leq \frac{16\alpha M\|z_0 - z^*\|^3}{T^{3/2}}.$$

*Let $M = 3\rho m$ and $\alpha = 2$. Algorithm 1 finds an $\epsilon$-saddle point within $\mathcal{O}(m^{2/3}\epsilon^{-2/3})$ iterations.*

*If we call the sub-procedure (Algorithm 5) with fixed $\eta^0_t = \bar{\eta}$, every call of this sub-procedure makes at most $\mathcal{O}(\log\log(\mathrm{poly}(m, \beta, \rho, \bar{\eta}, 1/\epsilon)))$ Newton steps.*

The above theorem shows that the CRN sub-problem can be solved to guarantee the desired precision for target problem in $\mathcal{O}(\log\log(1/\epsilon))$ iterations, which tightens the $\mathcal{O}(\log(1/\epsilon))$ iteration complexity in (Bullins & Lai, 2022; Adil et al., 2022). Additionally, (Bullins & Lai, 2022; Adil et al., 2022) requires additionally assume $\sigma_{\min}(\nabla \boldsymbol{F}(z)) \geq \mu$ for some positive constant $\mu$, which makes the problem similar to strongly-convex(-strongly-concave) problems, while our analysis does not require such an assumption.