# OpenReview forum: "Second-Order Min-Max Optimization with Lazy Hessians"
_ICLR.cc/2025/Conference — ICLR 2025 Oral_

### Official Review · Reviewer_LatJ · 2024-10-27

**Soundness:** 3
**Presentation:** 3
**Contribution:** 3
**Rating:** 6
**Confidence:** 4

**Summary:**

This paper proposes a second-order algorithm designed for convex-concave minimax optimization problems. The algorithm brings the Lazy Hessian (or Lazy *Jacobian* for this paper, I guess) approach by Doikov et al. (2023) to second-order minimax optimization algorithms as in Adil et al. (2022), Lin et al. (2022). The new algorithm can achieve a computational complexity smaller by an order of $d^{1/3}$ compared to previous results both for the convex-concave case. The paper also shows superlinear convergence in the strongly-convex-strongly-concave case (by adding a restarting scheme) where we also have an improvement of $d^{1/3}$.

**Strengths:**

- It is interesting to see that the idea of Doikov et al. (2023) can also work for minimax optimization problems.
- The paper seems to be well-written and clear, and there are empirical results supporting the theoretical arguments.

**Weaknesses:**

- The results only cover convex-concave minimax optimization for now, and there are no clear ways suggested to extend to broader problem classes like nonconvex-nonconcave functions or variational inequalities.
- As the main idea comes from combining two lines of previous work, it’s a bit hard to say that the ideas are highly original (except for the proof techniques of bounding the errors from lazy updates for the minimax case, particularly when there are EG steps in between).

**Questions:**

- In the proof of Theorem 4.1, what we essentially do is find an upper bound of
    $$ \begin{align*}
    \frac{1}{\sum_{i=0}^{t-1} \eta_i} \sum_{i=0}^{t-1} \eta_i \langle F(z_i),  z^{\star} - z_i\rangle
    \end{align*} $$
    which itself is an upper bound of the Gap function by Lemma A.1.

    My question is, can’t we just try to come up with an upper bound of the *MVI error* $\langle F(\hat{z}), z^{\star} - \hat{z} \rangle$ instead? If this works, it might be possible to extend the results to general MVIs. (I think one problem could be that we can’t use things like Jensen’s inequality if we step out of the convex-concave assumption.)

- Is the $\beta$ in Definition 3.4 just there for technical reasons (to ensure something like bounded iterates for the unconstrained case)?

- This might be a slightly irrelevant question, but any ideas of whether the proposed idea (or maybe the Lazy Hessian idea for minimization problems) could work for higher-order cases with $p \ge 3$?

- There seem to be complicated lower bound results in Adil et al. (2022) in terms of the *iteration* complexity. Can this naturally lead to some type of a lower bound in the *computational* complexity as well with which we can compare the new/previous upper bounds?

- Minor typo. In the middle of the inequality of Definition 3.3, I think it should be $f(x^{\star}, y^{\star})$.

**References.** \
Adil et al., 2022. Optimal methods for higher-order smooth monotone variational inequalities. \
Doikov et al., 2023. Second-order optimization with lazy Hessians. \
Lin et al., 2022. Explicit second-order min-max optimization methods with optimal convergence guarantee.

---

> ### Author Response · Authors · 2024-11-18
> **Response to Reviewer LatJ (in terms of "weakness")**
>
> We greatly appreciate the feedback from the reviewer. Below, we address the concerns of the reviewer.
>
> > W1: The results only cover convex-concave minimax optimization for now, and there are no clear ways suggested to extend to broader problem classes like nonconvex-nonconcave functions or variational inequalities.
>
> The convex-concave setting considered in our paper is foundamental and forms the basis of  broader problem classes like nonconvex-nonconcave functions or variational inequalities. And our result can be easily extended to several setups as the reviewer suggested.
>
> - Our results can be easily extended to monotone variational inequalities (MVI) just by simply changing the definition of gap function to ${\rm Gap}(\hat z; \beta)= \max_{z \in \mathbb{B}_{\beta}(z^*)} \langle F(z) , \hat z - z \rangle$ as in [1-3]. Then we can also upper bound the MVI error using Jensen's inequality like Lemma 2.7 in [3] to extend our result to MVIs.
> - Nonconvex-nonconcave functions are in general intractable [4], but our results can be extended to cohypomonotone nonconvex-nonconcave functions considered in some works  [5-9]. For this function class, if we define the saddle envelope (see e.g. Def. 5.3 [7], or Sec. 2 [8]), then by Proposition 5.2 [7], the saddle envelope is convex-concave and its gradient and Hessian are computable (e.g. see Lemma 2.1, 2.4 [8]). Therefore, we can simply run our algorithm on the convex-concave envelope in this setup.
>
> > W2: As the main idea comes from combining two lines of previous work, it’s a bit hard to say that the ideas are highly original (except for the proof techniques of bounding the errors from lazy updates for the minimax case, particularly when there are EG steps in between).
>
> The idea of reusing Hessian is proposed in 1960s [12]. Doikov et al. showed the advantage of reusing Hessian on minimization problem. Compared to their work, we consider (S)C-(S)C minimax optimization, which is more general and difficult than the (strongly)-convex minimization problem.
>
> Moreover, we have the following contributions compared with Doikov et al..
> - **We provide better complexity even for the strongly convex minimization.** Our results in Section 4.2 can be applied to solve strongly convex minimization problem as considered in Doikov et al. Doikov et al. gives the iteration complexity of $\tilde{\mathcal{O}}(m^2L_2^2/\mu^3)$ (Corollary D.4 in Doikov et al.). Our result of $\tilde{\mathcal{O}}(m^{2/3}L_2^{2/3}/\mu^{2/3})$ is strictly better than theirs.
> - **We provide novel insights on the optimality of the second-order methods.** Our results show that the optimal second-order methods in terms of the oracle complexity are not optimal in terms of the computation complexity. Doikov et al. finds the $(\epsilon,\sqrt{d\epsilon})$ stationary point for non-convex minimization problem which is weaker than the $(\epsilon,\sqrt{\epsilon})$ stationary point that the standard cubic regularized Newton method finds. Chayti et al. finds $f(x)-f^*\leq \epsilon$ with computational cost of $\mathcal{O}(d^{1/2}\epsilon^{-1/2}\cdot\texttt{Gradcost})$, whose dependency on $\epsilon$ cannot match the optimal second-order methods $\mathcal{O}(\epsilon^{-2/7})$ [1], and is strictly worse than the $\mathcal{O}(\epsilon^{-1/2}\cdot\texttt{Gradcost})$ of AGD (accelerated gradient descent [2]). In contrast, our computational complexity bound on $\epsilon$ matches the **optimal** second-order methods for minimax optimization, while reducing their dependency on $d$.
>
> In the revision, we also follow the suggestions of Reviewer 6zp1 to analyze an algorithm that  allows inexact auxiliary cubic regularized Newton (CRN) sub-problem solving (see Appendix H). In contrast, Doikov et al. supposes the CRN step is exactly computable. Additionally, we also design an efficient sub-procedure (Algorithm 5) to achieve the required accuracy of the auxiliary sub-problem in $O(\log \log (1/\epsilon))$ number of linear system solving. In contrast, [2-3]'s naive binary search procedure requires $O(\log (1/\epsilon))$ iterations, which is worse than ours.

---

> ### Author Response · Authors · 2024-11-18
> **References in our above responce**
>
> [1] Lin T, Jordan MI. Perseus: A simple and optimal high-order method for variational inequalities. Mathematical Programming. 2024.
>
> [2] Adil D, Bullins B, Jambulapati A, Sachdeva S. Optimal methods for higher-order smooth monotone variational inequalities. arXiv preprint arXiv:2205.06167. 2022.
>
> [3] Bullins B, Lai KA. Higher-order methods for convex-concave min-max optimization and monotone variational inequalities. SIAM Journal on Optimization. 2022.
>
> [4] Daskalakis C, Skoulakis S, Zampetakis M. The complexity of constrained min-max optimization. In STOC, 2021.
>
> [5] Diakonikolas J, Daskalakis C, Jordan MI. Efficient methods for structured nonconvex-nonconcave min-max optimization. In AISTATS, 2021.
>
> [6] Lee S, Kim D. Fast extra gradient methods for smooth structured nonconvex-nonconcave minimax problems. In NeurIPS. 2021
>
> [7] Chen L, Luo L. Near-optimal algorithms for making the gradient small in stochastic minimax optimization. arXiv preprint arXiv:2208.05925. 2022.
>
> [8] Grimmer B, Lu H, Worah P, Mirrokni V. The landscape of the proximal point method for nonconvex–nonconcave minimax optimization. Mathematical Programming. 2023.
>
> [9] Alacaoglu A, Kim D, Wright S. Revisiting Inexact Fixed-Point Iterations for Min-Max Problems: Stochasticity and Structured Nonconvexity. In ICML, 2024.
>
> [10] Monteiro RD, Svaiter BF. An accelerated hybrid proximal extragradient method for convex optimization and its implications to second-order methods. SIAM Journal on Optimization. 2013.
>
> [11] Nesterov Y. A method for solving the convex programming problem with convergence rate $\mathcal{O} (1/k^2)$. InDokl akad nauk Sssr, 1983.
>
> [12] VE Shamanskii. A modification of newton’s method. Ukrainian Mathematical Journal, 19(1): 118–122, 1967.

---

> ### Author Response · Authors · 2024-11-18
> **Response to Reviewer LatJ (in terms of "questions")**
>
> In terms of the questions, we address them below:
>
> > Q1: In the proof of Theorem 4.1, can’t we just try to come up with an upper bound of the MVI error instead? If this works, it might be possible to extend the results to general MVIs. (I think one problem could be that we can’t use things like Jensen’s inequality if we step out of the convex-concave assumption.)
>
> We have provided a positive answer to this question in our response to W1. Our results can be easily extended to monotone variational inequalities by simply modifying the gap function as we have discussed.
>
> > Q2: Is the $\beta$ in Definition 3.4 just there for technical reasons (to ensure something like bounded iterates for the unconstrained case)?
>
> Def 3.4 ensures the gap function is well-defined. If we do not restrict the gap function in a bounded domain, it can be undefined for simple convex-concave functions such as $f(x,y) = \langle x, y\rangle$.
>
> > Q3: This might be a slightly irrelevant question, but any ideas of whether the proposed idea (or maybe the Lazy Hessian idea for minimization problems) could work for higher-order cases with $p=3$?
>
> We think the idea of reusing Hessian can be applied to higher-order methods by reusing the $p$-th order derivatives and it will be an interesting future work.
>
> > Q4: There seem to be complicated lower bound results in Adil et al. (2022) in terms of the iteration complexity. Can this naturally lead to some type of a lower bound in the computational complexity as well with which we can compare the new/previous upper bounds?
>
> The lower bound in computational complexity is a very difficult unaddressed problem. Indeed, it remains open almost for any problems.  If we want to show computational lower bound for (second-order)optimization, we should first try to give lower bounds to basic linear algebra operations such as matrix multiplication.  However, to the best of our knowledge, even for matrix multiplication there only exists a trivial $\Omega(d^2)$ lower bound, which is the size of the output matrix.
>
> > Q5: In the middle of the inequality of Definition 3.3, I think it should be $f(x^*,y^*)$.
>
> Thanks for pointing it out. We have fixed the typo in the revision.

---

> > ### Author Response · Authors · 2024-11-23
> > **Response to Reviewer LatJ (cont.)**
> >
> > Thanks again for your valuable feedback! Could you please let us know whether your concerns have been addressed? We are happy to make further updates if you have any other questions or suggestions.

---

> > > ### Comment · Reviewer_LatJ · 2024-11-24
> > >
> > > I thank the authors for the detailed response, and sorry for the late reply.
> > >
> > > I am willing to raise my score to 7. (I am keeping my score for now because there seems to be no 7 in the score options right now.)

---

> > > > ### Author Response · Authors · 2024-11-24
> > > >
> > > > Thanks for your reply and the support to our work.

---

### Official Review · Reviewer_QgT2 · 2024-11-02

**Soundness:** 3
**Presentation:** 3
**Contribution:** 2
**Rating:** 8
**Confidence:** 3

**Summary:**

In this paper authors utilize the lazy Hessians technique inside second-order extragradient method for min-max optimization. The main idea of lazy Hessians technique is to update Hessian in second-order method only every $m$ iterations. Authors consider convex-concave and strongly-convex-strongly-concave setups. Authors claim, that if $m=\Theta(d)$ iterations, where $d$ is a dimension of the problem, their method achieves state-of-the-art computational cost.

**Strengths:**

Authors propose a method with the same oracle complexity, as state-of-the-art methods but with less computational complexity. This means, that their algorithm can achieve the same estimation error with the same number of iterations, but overall spending less computational resources and taking less time. The experimental results only support this point. This makes method more attractive from the practical point of view. The paper is written in a clear way, and it is easy to understand.

**Weaknesses:**

Overall, the paper feels like a very incremental result. Authors employ a known technique to reduce number of Hessian computations to existing second-order method to solve convex-concave min-max problem. To adapt proposed method to strongly-convex-strongly-concave problem, authors use a universal restarts framework, that works like a "wrap" around any method for convex(-concave) problems and gives better theoretical convergence for strongly-convex(-strongly-concave). Despite the fact that this paper proposes a new method with better computational complexity compared to other analogs, it lacks any novel ideas or solution of any complex problems.

## Typos
1. Lines 785-786: it should be $\frac{m^2 + 2m}{2}$
1. Lines 862-863: upper index in the second sum should be $\pi(t) + s  - 1$
2. Definition 3.5: in the end it should be $||\hat z - z^*||\le \epsilon$

**Questions:**

1. Could you explain please, how did you get second inequality in eq. (17)?
1. Lines 155-160. Sentence "Their methods only take a lazy CRN update (2) at each iteration..." seems to me very weird. I needed to read it couple of times to understand. Please, rewrite in in more clear way.

---

> ### Author Response · Authors · 2024-11-18
> **Response to Reviewer QgT2**
>
> We greatly appreciate the feedback from the reviewer. Below, we address the concerns of the reviewer.
>
> ### **Weakness**
>
> We kindly argue that the statement "this paper feels like a very incremental result" is not proper. Indeed, the idea of reusing Hessian is proposed in 1960s [3]. Doikov et al. showed the advantage of reusing Hessian on **minimization** problem. Compared to their work, we consider (S)C-(S)C **minimax** optimization, which is more general and difficult than the (strongly)-convex minimization problem. Employing such an idea to min-max problem and obtaining a better computation complexity itself requires new techniques in analysis such as (a) controlling the error of lazy Hessian led by the extra gradient step; (b) designing difficult subproblem to solve due to the absence of symmetric of $\nabla F(z)$.
>
> Our results are also not incremental as we have the following contributions compared with Doikov et al..
>
> - **We provide better complexity even for the strongly convex minimization.** Our results can in Section 4.2 be applied to solve strongly convex minimization problem as considered in Doikov et al. Doikov et al. gives the iteration complexity of $\tilde{\mathcal{O}}(m^2L_2^2/\mu^3)$ (Corollary D.4 in Doikov et al.). Our result of $\tilde{\mathcal{O}}(m^{2/3}L_2^{2/3}/\mu^{2/3})$ is strictly better than theirs.
> - **We provide novel insights on the optimality of the second-order methods.** Our results show that the optimal second-order methods in terms of the oracle complexity are not optimal in terms of the computation complexity. Doikov et al. finds the $(\epsilon,\sqrt{d\epsilon})$ stationary point for non-convex minimization problem which is weaker than the $(\epsilon,\sqrt{\epsilon})$ stationary point that the standard cubic regularized Newton method finds. Chayti et al. finds $f(x)-f^*\leq \epsilon$ with computational cost of $\mathcal{O}(d^{1/2}\epsilon^{-1/2}\cdot\texttt{Gradcost})$, whose dependency on $\epsilon$ cannot match the optimal second-order methods $\mathcal{O}(\epsilon^{-2/7})$ [1], and is strictly worse than the $\mathcal{O}(\epsilon^{-1/2}\cdot\texttt{Gradcost})$ of AGD (accelerated gradient descent [2]). In contrast, our computational complexity bound on $\epsilon$ matches the **optimal** second-order methods for minimax optimization, while reducing their dependency on $d$.
>
> In the revision, we also follow the suggestions of Reviewer 6zp1 to analyze an algorithm that  allows inexact auxiliary cubic regularized Newton (CRN) sub-problem solving.  We design an efficient sub-procedure (Algorithm 5) to solve the auxiliary sub-problem to the desired accuracy in only $\mathcal{O}(\log \log (1/\epsilon))$ number of linear system solving. In contrast, Doikov et al. supposes the CRN step is exactly computable and our results of the inexact algorithm may be of independent interest.
>
> In terms of the **typos**, we appreciate your detailed review and have fixed them in revision.
>
> [1] Monteiro RD, Svaiter BF. An accelerated hybrid proximal extragradient method for convex optimization and its implications to second-order methods. SIAM Journal on Optimization. 2013.
> [2] Nesterov Y. A method for solving the convex programming problem with convergence rate $\mathcal{O} (1/k^2)$. InDokl akad nauk Sssr, 1983.
> [3] VE Shamanskii. A modification of newton’s method. Ukrainian Mathematical Journal, 19(1): 118–122, 1967.
>
> ### **Questions**
>
> In terms of the **questions**, we address them below:
>
> > Q1: Could you explain please, how did you get second inequality in eq. (17)?
>
> Let $\beta = \Vert z_0 -  z^* \Vert$. We take maximum over  $z \in \mathbb{B}\_{3 \beta}(z^*)$
> in the inequality of Lemma A.1, and then use eq. (15) to obtain
>
> $$
> {\rm Gap}(\bar x_T, \bar y_T; 3\beta) \le \frac{ \max_{z \in \mathbb{B}\_{3 \beta}(z^*)}\Vert z_0 - z \Vert^2 }{2 \sum_{t=0}^{T-1} \eta_t} \le \frac{ 16 \Vert z_0 - z^* \Vert^2 }{2 \sum_{t=0}^{T-1} \eta_t},
> $$
>
> where the last step uses $\Vert z_0 - z \Vert \le \Vert z_0 - z^* \Vert + \Vert z - z^* \Vert \le 4 \beta$ for any $z \in \mathbb{B}_{3 \beta}(z^*)$.
>
> We apologize that we missed the notation "$\max_{z \in \mathbb{B}_{3 \beta}(z^*)}$" in our submitted version. We have also fixed this typo in the revision.
>
> > Q2: Lines 155-160. Sentence "Their methods only take a lazy CRN update (2) at each iteration..." seems to me very weird. I needed to read it couple of times to understand. Please, rewrite in in more clear way.
>
> We are grateful for the suggestions from the reviewer. In the revision, we have rewritten the sentence to: Their methods only take a lazy CRN update at each iteration, which makes it easy to bound the error of lazy Hessian updates using the Hessian Lipschitz assumption and the triangle inequality in the following way:
> \begin{align*}
> \Vert \nabla F(z_t)-\nabla F(z_{\pi(t)}) \Vert \leq \rho \Vert  z_{\pi(t)}-z_t \Vert  \leq \rho \sum_{i=\pi(t)}^{t-1} \Vert  z_{i}-z_{i+1} \Vert .
> \end{align*}

---

> > ### Author Response · Authors · 2024-11-23
> > **Response to Reviewer QgT2 (cont.)**
> >
> > Thanks again for your valuable feedback! Could you please let us know whether your concerns have been addressed? We are happy to make further updates if you have any other questions or suggestions.

---

> ### Comment · Reviewer_QgT2 · 2024-11-24
>
> I thank the authors for such an answer and the further improvement of the paper. I've got a minor question.
> ### Q1
> Am I correct, that you do not just take $\max_{z \in B_{3\beta}(z^*)}$ in Lemma A.1, but take $\min_{x \in B_{3\beta}(x^*)} \max_{y \in B_{3\beta}(y^*)}$ to get duality gap, and then you do $\min_{x \in B_{3\beta}(x^*)} \max_{y \in B_{3\beta}(y^*)} ||z_0 - z||^2 \le \max_{z \in B_{3 \beta}(z^*)} ||z_0 - z||^2$, due to convexity of squared norm?

---

> ### Author Response · Authors · 2024-11-24
> **Response to Reviewer QgT2 (cont.)**
>
> Thanks for your prompt response and engagement in the discussion!
>
> In terms of Q1, our proof uses the following relationship:
> $$
> {\rm Gap}(\bar x_T, \bar y_T ; 3 \beta) = \max_{y \in B_{3 \beta}(y^*)} f(\bar x_T,y) - \min_{x \in B_{3 \beta} (x^*) }f(x, \bar y_T) =
> \max_{x \in B_{3 \beta}(x^*), y \in B_{3 \beta}(y^*)} \\{ f(\bar x_T, y) - f(x, \bar y_T) \\}  \le \max_{z \in B_{3 \sqrt{2} \beta}(z^*)} \\{ f(\bar x_T, y) - f(x, \bar y_T) \\},
> $$
> and then applies Lemma A.1. We should say "take maximum over $z \in B_{3 \sqrt{2} \beta}(z^*)$" instead of "take maximum over $z \in B_{3 \beta}(z^*)$".  However, since it only differs by a constant factor of $\sqrt{2}$, it does not affect our result. We really appreciate your careful reading that helped us identify our typo. We are very glad to answer any other questions you may have.

---

> > ### Comment · Reviewer_QgT2 · 2024-11-24
> >
> > Sorry, but it seems to me that you have a mistake here, since in the second equality you say $\min_{x \in B_{3\beta}(x^*)} f(x, \bar y_T) = \max_{x \in B_{3\beta}(x^*)} f(x, \bar y_T)$.

---

> ### Author Response · Authors · 2024-11-24
>
> Thanks for your reply. We use the fact that $-\min_{x \in B_{3\beta}(x^*)} f(x, \bar y_T) = \max_{x \in B_{3\beta}(x^*)}(- f(x, \bar y_T))$ in the second equality.
> We hope this can address your concern. Please also let us know if you have any further questions and we are very glad to answer them.

---

> > ### Author Response · Authors · 2024-11-26
> >
> > Dear Reviewer QgT2,
> >
> > We hope our responses answer all your questions!
> >
> > In case you need any remaining clarifications, we would be more than happy to reply. If your questions are all properly addressed, we really hope that you consider increasing your score to support our work.
> >
> > Regards,
> > Authors

---

> > > ### Comment · Reviewer_QgT2 · 2024-11-26
> > >
> > > I thank the reviewers for addressing my concerns and answering my questions. Due to this and the fact that authors additionally introduce methodology to solve inner subproblem, I increase my score to 8 but decrease confidence level to 3, because I did not carefully check the proofs of CRN subproblem solving.

---

> > > > ### Author Response · Authors · 2024-11-26
> > > >
> > > > Thanks for your support to our work. We sincerely appreciate your recognition!

---

### Official Review · Reviewer_6zp1 · 2024-11-02

**Soundness:** 3
**Presentation:** 3
**Contribution:** 3
**Rating:** 8
**Confidence:** 4

**Summary:**

Authors generalize the results of Nikita Doikov, El Mahdi Chayti, and Martin Jaggi. Second-order optimization with lazy hessians. In ICML, 2023. related with lazy Hessian for Cubic regularized Newton method for Saddle-point problems (SPP). The developed result was cleary presented and sufficiently technical. Moreover, requires principally new ideas. However, the the motivation to consider second-order methods are quite limited and is mainly theoretical from my point of view. Also in the paper authors doesn't observe the accuracy required to solve auxiliary problem.

**Strengths:**

I guess the paper is good from mathematical point of view. The results is strong, original, well presented. I agree with authors that they developed significantly new tricks to work with this class of problems. I guess the paper is good!

**Weaknesses:**

For me the main drawback is motivation. I do not understand why we should use second-order method with expensive iteration rather than the first-order one. I understand the motivation for convex optimization where the number of iteration significantly reduces by using optimal second-order scheme, but I do not understand it for SPP where the difference is minor.

**Questions:**

1) Is it possible to generalize the result in case strongly convex-strongly concave case with different constant of strong convexity/concavity?
2) Could you estimate the required accuracy for auxiliary problem that guarantee the desired precision for target problem?
 I can make rating higher if you can positively answer for this questions.

---

> ### Author Response · Authors · 2024-11-18
> **Response to Reviewer 6zp1**
>
> We greatly appreciate the feedback from the reviewer. Below, we address the concerns of the reviewer.
>
> ### **Weakness**
>
> For convex optimization, the optimal rates of first-order methods are $\mathcal{O}(\epsilon^{-\frac{1}{2}})$ [1], while for second-order methods are $\mathcal{O}(\epsilon^{-\frac{2}{7}})$ [2]. The gap of convergence rate in first-order and second-order methods is $\mathcal{O}(\epsilon^{-\frac{3}{14}})$.
>
> For saddle point problems (SPP), the second-prder methods can achieve the convergence of $O(\epsilon^{-2/3})$, which is better than the $O(\epsilon^{-1})$ optimal rates of first-order methods. The gap of convergence rate in first-order and second-order methods is $\mathcal{O}(\epsilon^{-\frac{1}{3}})$, which is **even larger** than the one for convex optimization.
>
> Hence, we do not think the difference in SPP is minor and we think there exists sufficient motivation for designing efficient second-order methods for SPP just like the motivation for convex optimization.
>
> ### **Questions**
>
> In terms of the **questions**, we address them below:
>
> > **Questions 1**: Is it possible to generalize the result in case strongly convex-strongly concave case with different constant of strong convexity/concavity?
>
> For first-order algorithmsm, we can use the Catalyst acceleration to achieve near-optimal rates for (S)C-(S)C case with different constant of strong convexity/concavity [3]. It is open whether one can develop a second-order Catalyst and it will be an interesting future work.
>
> > **Questions 2**: Could you estimate the required accuracy for auxiliary problem that guarantee the desired precision for target problem? I can make rating higher if you can positively answer for this questions.
>
> We can simply modify our algorithm to an inexact version as follows. Recall our algorithm:
> $$
> \begin{align*}
> z_{t+1/2} = z_t - ( \nabla F(z_{\pi(t)}) +  \gamma_t I_d )^{-1} F(z_t), \quad \quad \\
> z_{t+1} = z_t - \gamma_t^{-1} F(z_{t+1/2})
> \end{align*}
> $$
> Our exact algorithm means taking
> $$
> \gamma_t = M \Vert z_t - z_{t+1/2} \Vert~~~~~~~~~~~~(1)
> $$
> in the above equation. Similar to the case in [5], we can get the same convergence rate (ignoring the constant factor) if condition (1) is weaken to
> $$
> M \Vert z_t - z_{t+1/2} \Vert \le \gamma_t \le 2 M \Vert z_t - z_{t+1/2} \Vert.
> $$
> We only need a few lines of modifications (see the purple text in Section D) to the proof of the exact algorithm to give the proof for the inexact algorithm. We present a self-contained section for the inexact algorithm in Appendix H, and we also design an efficient sub-procedure (Algorithm 5) to achieve the required accuracy of the auxiliary sub-problem to desired accuracy in only $\mathcal{O}(\log \log (1/\epsilon))$ number of linear system solving, which improves the $\mathcal{O}(\log (1/\epsilon))$ complexity of [4,5].
>
> We thank the reviewer for his/her suggestion on the inexact algorithm. We are happy to make further updates if the reviewer has any other questions or suggestions.
>
> [1] Nesterov Y. Lectures on convex optimization. Berlin: Springer; 2018.
>
> [2]  Arjevani Y, Shamir O, Shiff R. Oracle complexity of second-order methods for smooth convex optimization. Mathematical Programming. 2019.
>
> [3] Lin T, Jin C, Jordan MI. Near-optimal algorithms for minimax optimization. In COLT, 2020.
>
> [4] Adil D, Bullins B, Jambulapati A, Sachdeva S. Optimal methods for higher-order smooth monotone variational inequalities. arXiv preprint arXiv:2205.06167. 2022.
>
> [5] Bullins B, Lai KA. Higher-order methods for convex-concave min-max optimization and monotone variational inequalities. SIAM Journal on Optimization. 2022.

---

> > ### Author Response · Authors · 2024-11-23
> > **Response to Reviewer 6zp1 (cont.)**
> >
> > Thanks again for your valuable feedback! Could you please let us know whether your concerns have been addressed? We are happy to make further updates if you have any other questions or suggestions.

---

> > ### Comment · Reviewer_6zp1 · 2024-11-23
> >
> > I understand the argument $\mathcal{O}(\epsilon^{-\frac{3}{14}})$ vs $\mathcal{O}(\epsilon^{-\frac{1}{3}})$, but it seems that here it's worth to use another criteria, related with relative acceleration in terms $1/k^2 \to 1/k^{7/2}$ (win 1/k^{3/2}) and $1/k \to 1/k^{3/2}$ (win 1/k^{1/2}).
> >
> > I'm satisfied about Question 2 and plan to raise my score to 7.

---

> > > ### Author Response · Authors · 2024-11-24
> > > **Response to Reviewer 6zp1 (cont.)**
> > >
> > > Thanks for your prompt response and engagement in the discussion, and your willingness to raise your score.
> > >
> > > We agree that the criterion you have mentioned is also a valid perspective. However, our criterion, which compares the dependency on $\epsilon$, can tell us when to use first-order or second-order methods.
> > > For instance, for minimization problems, optimal first-order and second-order methods have the complexity of $\mathcal{O}( d \epsilon^{-1/2})$ and $\mathcal{O}(d^3 \epsilon^{-\frac{2}{7}})$, respectively. Therefore, second-order methods outperform first-order methods when $d \lesssim \epsilon^{-\frac{3}{28}}$.
> > > For min-max optimization, optimal first-order and second-order methods have the complexity of $\mathcal{O}( d \epsilon^{-1})$ and $\mathcal{O}(d^3 \epsilon^{-\frac{2}{3}})$, respectively, which suggests that second-order methods outperform first-order methods when $d \lesssim \epsilon^{-\frac{1}{6}}$. Due to the above analysis, given $\epsilon$ and $d$, the region where second-order methods outperform first-order methods in min-max optimization is **larger** than in minimization problems.
> > >
> > > Our methods take the complexity of $\mathcal{O}(d^{3}+d^{8/3}\epsilon^{-2/3})$, which outperforms the first-order methods when $d\lesssim \epsilon^{-1/5}$, which further enlarge such region. In the above analysis, we take $N=d$ for simplification, where $N$ is the computational cost for the first-order oracle.
> > >
> > > If you think our responce address your concern, we kindly hope you can consider updating your assessment to support our work.

---

> > > > ### Comment · Reviewer_6zp1 · 2024-11-24
> > > >
> > > > I'm not sure about the complexity $O(d\epsilon^{-1/2})$ in comparison with  $O(d^3\epsilon^{-2/7})$. For me the complexity of the first order-method should be $O(d^2\epsilon^{-1/2})$ in comparison with  $O(d^3\epsilon^{-2/7})$. Indeed, if you consider f(Ax) this is typically situation.

---

> ### Author Response · Authors · 2024-11-24
>
> Thanks for your reply.
>
> If we take the complexity of the first-order oracles as $\mathcal{O}(d^2)$ for min and min-max problems,  the situation for the motivation of considering min- and min-max problems remains the same. We list the trade-off between $\epsilon^{-1}$ and $d$ for first and second order methods again:
>
> * For the minimization problems, optimal second-order methods outperform optimal first-order methods when $d \lesssim \epsilon^{-\frac{3}{14}}$ ($d^3\epsilon^{-2/7}\lesssim d^2\epsilon^{-1/2}$).
>
> * For the min-max problems,  optimal second-order methods outperform optimal first-order methods when $d \lesssim \epsilon^{-\frac{1}{3}}$ ($d^3\epsilon^{-2/3}\lesssim d^2\epsilon^{-1}$). The region where second-order methods outperform first-order methods in min-max optimization is still larger than in minimization problems.
>
> * For our methods for min-max problems, it outperforms optimal first-order methods when $d \lesssim \epsilon^{-1/2}$ ($d^3 + d^{8/3}\epsilon^{-2/3}\lesssim d^2\epsilon^{-1}$), which further enlarges the region.
>
> We agree with you that we can also use $k^{-\alpha}$ as the criteria, but we kindly disagree that the choice of the criteria impact the motivation nor the contribution of this work. Again, we sincerely appreciate your helpful review and the support to our work!

---

### Official Review · Reviewer_Z87Q · 2024-11-04

**Soundness:** 3
**Presentation:** 3
**Contribution:** 3
**Rating:** 8
**Confidence:** 4

**Summary:**

This work proposed second-order algorithms LEN and LEN-restart for C-C and SC-SC min-max problems, these algorithms incorporate lazy Hessian update and Extragradient update, the computational cost analysis and numerical experiment results showed the outperformance of the proposed algorithms versus existing algorithms.

**Strengths:**

1. A new algorithm in min-max optimization with better computational complexity versus existing results.
2. The paper is well organized, the flow is easy to follow.

**Weaknesses:**

1. The main component seems to be a combination of Doikov et al. (2023) on lazy Hessian and Adil et al., (2022) on extragradient, which may restrict the novelty a bit.
2. The experiment can be further enhanced.
   - First, the $O(d^{1/3})$ improvement suggests the outperformance is valid in high-dimensional cases (while not in low-dimensional cases), now the experiment cannot exhibit such a pattern, how does the algorithm perform in low-dimensional case?
   - It is not clear how the choice of $m$ affects the performance, maybe you can follow Doikov et al. (2023) to add more experiments for clarification.
   - The theory part uses the gap function as the measurement, while the experiment uses the gradient norm or point distance, maybe you can add more clarification to rationalize your choice.

**Questions:**

/

---

> ### Author Response · Authors · 2024-11-18
> **Response to Reviewer Z87Q (in terms of "weakness 1")**
>
> We greatly appreciate the feedback from the reviewer. Below, we address the concerns of the reviewer on **weakness 1 "The main component seems to be a combination of Doikov et al. on lazy Hessian and Adil et al. on extragradient, which may restrict the novelty a bit."**
>
> In terms of novelty, we are indeed inspired by the Doikov et al., as we discussed in the manuscript. However, we kindly argue that the combination of Doikov et al. on lazy Hessian and Adil et al. on extragradient is non-trivial as we need to control the error of lazy Hessian led by the extra gradient step.
>
> Moreover, we have the following contributions compared with Doikov et al..
> - **We provide better complexity even for the strongly convex minimization.** Our results in Section 4.2 can be applied to solve strongly convex minimization problem as considered in Doikov et al. Doikov et al. gives the iteration complexity of $\tilde{\mathcal{O}}(m^2L_2^2/\mu^3)$ (Corollary D.4 in Doikov et al.). Our result of $\tilde{\mathcal{O}}(m^{2/3}L_2^{2/3}/\mu^{2/3})$ is strictly better than theirs.
> - **We provide novel insights on the optimality of the second-order methods.** Our results show that the optimal second-order methods in terms of the oracle complexity are not optimal in terms of the computation complexity. Doikov et al. finds the $(\epsilon,\sqrt{d\epsilon})$ stationary point for non-convex minimization problem which is weaker than the $(\epsilon,\sqrt{\epsilon})$ stationary point that the standard cubic regularized Newton method finds. Chayti et al. finds $f(x)-f^*\leq \epsilon$ with computational cost of $\mathcal{O}(d^{1/2}\epsilon^{-1/2}\cdot\texttt{Gradcost})$, whose dependency on $\epsilon$ cannot match the optimal second-order methods $\mathcal{O}(\epsilon^{-2/7})$ [1], and is strictly worse than the $\mathcal{O}(\epsilon^{-1/2}\cdot\texttt{Gradcost})$ of AGD (accelerated gradient descent [2]). In contrast, our computational complexity bound on $\epsilon$ matches the **optimal** second-order methods for minimax optimization, while reducing their dependency on $d$.
>
> In the revision, we also follow the suggestions of Reviewer 6zp1 to analyze an algorithm that allows inexact auxiliary cubic regularized Newton (CRN) sub-problem solving (see Appendix H). In contrast, Doikov et al. supposes the CRN step is exactly computable. Additionally, we also design an efficient sub-procedure (Algorithm 5) to to achieve the required accuracy of the auxiliary sub-problem in $O(\log \log (1/\epsilon))$ number of linear system solving. In contrast, Adil et al.'s binary search procedure requires $O(\log (1/\epsilon))$ iterations, which is worse than ours.
>
> [1] Monteiro RD, Svaiter BF. An accelerated hybrid proximal extragradient method for convex optimization and its implications to second-order methods. SIAM Journal on Optimization. 2013.
> [2] Nesterov Y. A method for solving the convex programming problem with convergence rate $\mathcal{O} (1/k^2)$. In Dokl akad nauk Sssr, 1983.

---

> ### Author Response · Authors · 2024-11-18
> **Response to Reviewer Z87Q (in terms of "weakness 2")**
>
> We greatly thank the reviewer for his/her detailed suggestions on our experiments, and we have enhanced the experiments following the suggestions.
>
> > 1. First, the theory suggests the outperformance is valid in high-dimensional cases (while not in low-dimensional cases), now the experiment cannot exhibit such a pattern, how does the algorithm perform in low-dimensional case?
>
> We add an additional experiment in low dimension $(d=10)$ in the revision. See Figure 1(a). In such case, the effect of using and not using the lazy Hessian is quite similar.
>
> > 2. It is not clear how the choice of $m$ affects the performance, maybe you can follow Doikov et al. (2023) to add more experiments for clarification.
>
> We present the impact of different values of $m$ on the algorithm performance in Figure 1.
>
> > 3. The theory part uses the gap function as the measurement, while the experiment uses the gradient norm or point distance, maybe you can add more clarification to rationalize your choice.
>
> Although the theory part uses the gap function as the measurement, the duality gap is difficult to compute in practice. Therefore, we use the gradient norm as an alternative measurement.  The gradient norm is also a valid measurement because a small gradient norm can imply a small gap due to the Cauchy–Schwarz inequality $\langle F(z),z-z^*\rangle \le \Vert F(z) \Vert \Vert z - z^* \Vert$. Below, we list some works that also use the duality gap in theory but use gradient norm in experiments [2-4].
>
> The convergence of point distance $\Vert z - z^* \Vert$ for convex(-concave) optimization is impossible in general due to the lower bound (see the second inequality in  Theorem 2.1.7 [1]). However, we observe the convergence on Problem (11), which indicates that this problem has a special structure. We think it is an interesting finding and thus also present it in our paper.
>
> [1] Nesterov Y. Lectures on convex optimization. Berlin: Springer; 2018.
>
> [2] Alves MM, Svaiter BF. A search-free $\mathcal{O}(1/k^{3/2})$ homotopy inexact proximal-Newton extragradient algorithm for monotone variational inequalities. arXiv preprint arXiv:2308.05887. 2023.
>
> [3] Kevin Huang, Shuzhong Zhang. An Approximation-Based Regularized Extra-Gradient Method for Monotone Variational Inequalities. arXiv 2210.04440. 2022.
>
> [4] Jiang R, Kavis A, Jin Q, Sanghavi S, Mokhtari A. Adaptive and Optimal Second-order Optimistic Methods for Minimax Optimization. In NeurIPS, 2024.

---

> > ### Author Response · Authors · 2024-11-23
> > **Response to Reviewer Z87Q (cont.)**
> >
> > Thanks again for your valuable feedback! Could you please let us know whether your concerns have been addressed? We are happy to make further updates if you have any other questions or suggestions.

---

> > > ### Comment · Reviewer_Z87Q · 2024-11-24
> > >
> > > I am satisfied with the reply, and will raise the score. Thank you.

---

> > > > ### Author Response · Authors · 2024-11-24
> > > > **Response to Reviewer Z87Q (cont.)**
> > > >
> > > > Thanks for your reply. We sincerely appreciate your recognition of our work!

---

### Author Response · Authors · 2024-11-20
**Global Response to all the Reviewers**

We thank all the reviewers for their valuable and constructive comments. Overall, we appreciate that all the reviewers acknowledged our contributions in proving a better computational complexity for second-order min-max optimization.

One common concern of the reviewers is the comparison with Doikov et al. which analyzes lazy Hessian method in minimzation problems. We highlight the following novelty and contributions of our work compared with Doikov et al..

- **We provide better complexity even for the strongly convex minimization.** Our results can in Section 4.2 be applied to solve strongly convex minimization problem as considered in Doikov et al. Doikov et al. gives the iteration complexity of $\tilde{\mathcal{O}}(m^2L_2^2/\mu^3)$ (Corollary D.4 in Doikov et al.). Our result of $\tilde{\mathcal{O}}(m^{2/3}L_2^{2/3}/\mu^{2/3})$ is strictly better than theirs.
- **We provide novel insights on the optimality of the second-order methods.** Our results show that the optimal second-order methods in terms of the oracle complexity are not optimal in terms of the computation complexity. Doikov et al. finds the $(\epsilon,\sqrt{d\epsilon})$ stationary point for non-convex minimization problem which is weaker than the $(\epsilon,\sqrt{\epsilon})$ stationary point that the standard cubic regularized Newton method finds. Chayti et al. finds $f(x)-f^*\leq \epsilon$ with computational cost of $\mathcal{O}(d^{1/2}\epsilon^{-1/2}\cdot\texttt{Gradcost})$, whose dependency on $\epsilon$ cannot match the optimal second-order methods $\mathcal{O}(\epsilon^{-2/7})$ [1], and is strictly worse than the $\mathcal{O}(\epsilon^{-1/2}\cdot\texttt{Gradcost})$ of AGD (accelerated gradient descent [2]). In contrast, our computational complexity bound on $\epsilon$ matches the **optimal** second-order methods for minimax optimization, while reducing their dependency on $d$.

Moreover, we have enhanced our paper by following the suggestions from the reviewers.

- **We have analyzed an algorithm that allows inexact auxiliary cubic regularized Newton (CRN) sub-problem solving following the suggestions from Reviewer 6zp1  (see the new Appendix H) .**
We show that the desired accuracy of the sub-problem is dynamic such that
$$
M \Vert z_t - z_{t+1/2} \Vert \le \gamma_t \le 2 M \Vert z_t - z_{t+1/2} \Vert, ~~~~~~~ \text{(see Algorithm 4)}
$$
which can be satisfied within $\mathcal{O}(\log \log (1/\epsilon))$ numbers of linear system solving (the linear system  can be solved within $\mathcal{O}(d^2)$ time, please refer to line 378-line 397 in Section 4.3). In contrast, Doikov et al. supposes the CRN step is exactly computable. **Additionally**, we also design an efficient sub-procedure (Algorithm 5) to achieve the required accuracy of the auxiliary sub-problem in only $\mathcal{O}(\log \log (1/\epsilon))$ number of linear system solving. In contrast, [3-4]'s naive binary search procedure requires $\mathcal{O}(\log (1/\epsilon))$ iterations, which is worse than ours.

- **We have enhanced the experiments following the suggestions from Reviewer Z87Q (see the updated Figure 1).** We have included additional experiments in low dimension $(d=10)$ and ablation study on the effect of hyperparamter $m$.

We have also addressed all other issues raised by the reviewers. We hope that the reviewers will engage with us in a back-and-forth discussion and we will be very happy to answer any further questions.

[1] Monteiro RD, Svaiter BF. An accelerated hybrid proximal extragradient method for convex optimization and its implications to second-order methods. SIAM Journal on Optimization. 2013.

[2] Nesterov Y. A method for solving the convex programming problem with convergence rate $\mathcal{O} (1/k^2)$. InDokl akad nauk Sssr, 1983.

[3] Adil D, Bullins B, Jambulapati A, Sachdeva S. Optimal methods for higher-order smooth monotone variational inequalities. arXiv preprint arXiv:2205.06167. 2022.

[4] Bullins B, Lai KA. Higher-order methods for convex-concave min-max optimization and monotone variational inequalities. SIAM Journal on Optimization. 2022.

---

### Meta-Review · Area_Chair_znrm · 2024-12-14

**Metareview:**

This paper studies second-order methods for convex-concave minimax problem. In particular, the authors showed that the complexity result of previous work can be improved by reusing the Hessian information, which is called the lazy Hessians technique. Overall, the paper presents very novel results with better complexity and novel insights for second-order methods for minimax problem. This is of great importance to both optimization and machine learning community.

**Additional Comments On Reviewer Discussion:**

Reviewers all agreed that this is a good paper.

---

### Decision · Program_Chairs · 2025-01-22

Accept (Oral)